# Annual cycle observations of aerosols capable of ice formation in central Arctic clouds

Jessie M. Creamean [1✉], Kevin Barry[1], Thomas C. J. Hill [1], Carson Hume[1], Paul J. DeMott [1], Matthew D. Shupe [2,3], Sandro Dahlke[4], Sascha Willmes[5], Julia Schmale [6], Ivo Beck[6], Clara J. M. Hoppe [7], Allison Fong[7], Emelia Chamberlain [8], Jeff Bowman [8], Randall Scharien[9] & Ola Persson[2,3]

The Arctic is warming faster than anywhere else on Earth, prompting glacial melt, permafrost thaw, and sea ice decline. These severe consequences induce feedbacks that contribute to amplified warming, affecting weather and climate globally. Aerosols and clouds play a critical role in regulating radiation reaching the Arctic surface. However, the magnitude of their effects is not adequately quantified, especially in the central Arctic where they impact the energy balance over the sea ice. Specifically, aerosols called ice nucleating particles (INPs) remain understudied yet are necessary for cloud ice production and subsequent changes in cloud lifetime, radiative effects, and precipitation. Here, we report observations of INPs in the central Arctic over a full year, spanning the entire sea ice growth and decline cycle. Further, these observations are size-resolved, affording valuable information on INP sources. Our results reveal a strong seasonality of INPs, with lower concentrations in the winter and spring controlled by transport from lower latitudes, to enhanced concentrations of INPs during the summer melt, likely from marine biological production in local open waters. This comprehensive characterization of INPs will ultimately help inform cloud parameterizations in models of all scales.

[1] Department of Atmospheric Science, Colorado State University, Fort Collins, CO, USA. [2] Cooperative Institute for Research in Environmental Sciences, University of Colorado, Boulder, CO, USA. [3] Physical Sciences Laboratory, National Oceanic and Atmospheric Administration, Boulder, CO, USA. [4] Climate Sciences Division, Alfred-Wegener-Institut, Helmholtz-Zentrum für Polar- und Meeresforschung, Potsdam, Germany. [5] Environmental System Analysis and Modeling, Universität Trier, Trier, Germany. [6] Extreme Environments Research Laboratory, École Polytechnique Fédérale de Lausanne, Lausanne, Switzerland. [7] Biosciences Division, Alfred-Wegener-Institut, Helmholtz-Zentrum für Polar- und Meeresforschung, Bremerhaven, Germany. [8] Scripps Institution of Oceanography, University of California, San Diego, CA, USA. [9] Department of Geography, University of Victoria, Victoria, BC, Canada. ✉email: jessie.creamean@colostate.edu

Aerosols are an important component of the atmosphere through their impacts on climate, yet the magnitude of their effects remains unquantified[1]. The largest uncertainties stem from the indirect effects of aerosols on clouds, which are poorly represented in models[2]. This deficiency is especially true for the central Arctic Ocean, in part, due to a dearth of observations and understanding of clouds and aerosols collocated in space and time[3]. Evaluating aerosol-cloud interactions in this subregion is critical given the heightened sensitivity of the surface energy budget to cloud phase over sea ice.

While the enigmatic behavior of clouds causes an incomplete understanding of their role in the Arctic climate system, previous efforts agree on some general qualities. Arctic clouds are most frequent in autumn and least in winter[4]. Over the sea ice, they impart seasonally-contrasting radiative impacts, warming the surface for most of the year while cooling it for a short period in summer[5,6]. Arctic mixed-phase clouds (AMPCs) composed of liquid and ice are persistent[7] and common, occurring 41% of the time on average and up to 70% of the time during the spring and autumn transition seasons[8]. The ability for models to predict the sea ice seasonal cycle largely depends on accurate depictions of AMPCs because of their strong influence on the surface energy budget, but the magnitude of their impact on downwelling radiation inherently depends on the relative amounts of liquid vs. ice[5,9]. However, models struggle to represent seasonal variation in AMPCs without significant biases[4]. Further, inter-model differences in Arctic cloud occurrence throughout the year are driven by inadequate parameterizations, motivating the need for observations of fundamental microphysical processes[4,10].

Cloud formation requires the presence of aerosols[11]. At supercooled liquid temperatures (–38 to 0 °C), the primary pathway for cloud ice formation is immersion freezing by ice nucleating particles (INPs)[12], wherein aerosols are first immersed in cloud droplets followed by freezing once optimal conditions are met. The dominant types of Arctic INPs have been shown to comprise of mineral and biological materials, emanating from land (e.g., glacial dust[13], permafrost[14], vegetation[15]) and the ocean (e.g., sea spray[16], biological productivity[17,18]). At relatively warm temperatures (above –15 °C)—common among central Arctic clouds throughout the year[8]—the majority of INPs are typically biological[19]. Pioneering studies by Bigg et al. during the early 1990s alluded to marine biota as the source of central Arctic INPs[20,21]. While previous studies characterizing central Arctic INPs are salient, they are also rare, and typically confined to summer, aside from a single study from March[18].

Here, we describe a full annual cycle of size-resolved INPs in the central Arctic. To capture the breadth of INP sources and assess their impacts on Arctic clouds and the surface energy budget, a complete annual cycle of observations is essential. The Multidisciplinary drifting Observatory for the Study of Arctic Climate (MOSAiC) expedition offered the opportunity to achieve a holistic characterization of the central Arctic coupled system over the course of a full year (September 2019–October 2020)[22], specifically enabling a comprehensive investigation of atmospheric, oceanographic, and sea ice observations to characterize the sources of INPs. Our findings portray strong seasonality of INPs, with lower concentrations in the winter and spring controlled by transport from lower latitudes, to enhanced concentrations of INPs during the summer, likely from marine biological production in local open waters.

## Results and discussion

**Setting the stage: state of the sea ice, ocean, and atmosphere.** As context for our INP observations, MOSAiC measurements presented here were made onboard the research vessel Polarstern[23] as it drifted from the eastern Arctic Ocean, past the North Pole, and on toward Svalbard, mostly within the central Arctic pack ice (Figs. 1 and 2). Here, we define Arctic autumn, winter, spring, and summer as September–November, December–March, April–May, and June–August, respectively[22]. Open lead area fraction (i.e., cracks in the pack ice that expose open ocean) around Polarstern reached a maximum of 6% in early May, while melt pond fraction reached its maximum of 37% in early August. The concentration of sub-ice chlorophyll-*a* (chl-*a*)—a proxy for biomass of marine primary producers—was consistent with what has been previously reported for the central Arctic Ocean[24], peaking in June–July. The coldest air temperature (down to –42 °C) occurred toward the end of winter, while, following the melt onset in late May, the warmest (up to 3 °C) was observed during the 24-h sunlit summer when air and surface temperature hovered around 0 °C[22]. The summer experienced relatively calmer wind (4.5 ± 1.8 m s$^{-1}$ on average) compared to the rest of the year (5.9 ± 2.6 m s$^{-1}$ on average).

Airmass trajectory analyses provided a broader perspective for the conditions at Polarstern. Figure 3 shows the sea ice concentration (SIC) along the pathways of transport to Polarstern over the course of 3 days back ending at 30 m above mean sea level (AMSL) in time colored by average transport latitude, and corresponding average transport height. The same analysis for 5 and 7 days back in time, and ending at 100 m AMSL, is shown in the Supplementary Information and yielded similar results (Supplementary Figs. 1 and 7). We note that the trajectory analyses do not enable us to determine the exact origin of the airmasses, but, in tandem with complementary observations, provide insight into the possible source regions of the measured INPs. On average, 3-day airmasses mostly resided over the pack ice (70% of the time) with some passing over the marginal ice zone (MIZ; 27% of the time), and infrequently over ice-free open water or land (3% of the time, mostly during port transits). Extended periods throughout autumn and spring were subject to central Arctic transport from over pack ice, while episodic periods of transport from over the MIZ occurred in winter and summer. Height along the 3-day trajectories varied but typically remained below 500 m (87% of the

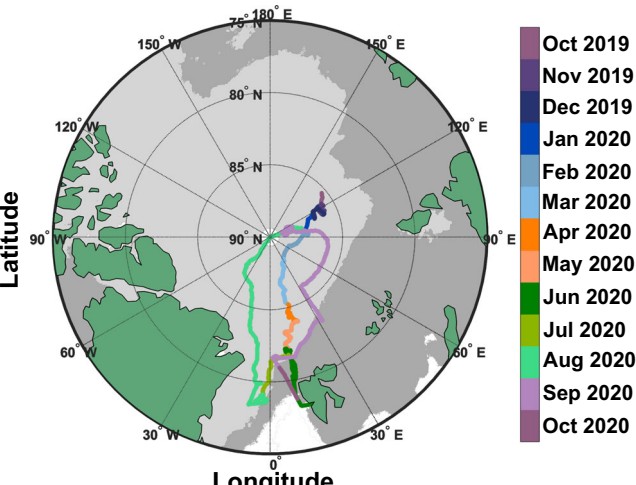

**Fig. 1 Map of Polarstern location during the Multidisciplinary drifting Observatory for the Study of Arctic Climate (MOSAiC) expedition.** The ship track is colored by date. Only dates that involved ice nucleating particle (INP) sample collection are shown. Dark and light gray shaded areas represent the maximum and minimum sea ice extent (sea ice concentration ≥15%) in March and September 2020, respectively. Maps were created by the authors using MathWorks MATLAB (https://www.mathworks.com/products/matlab.html).

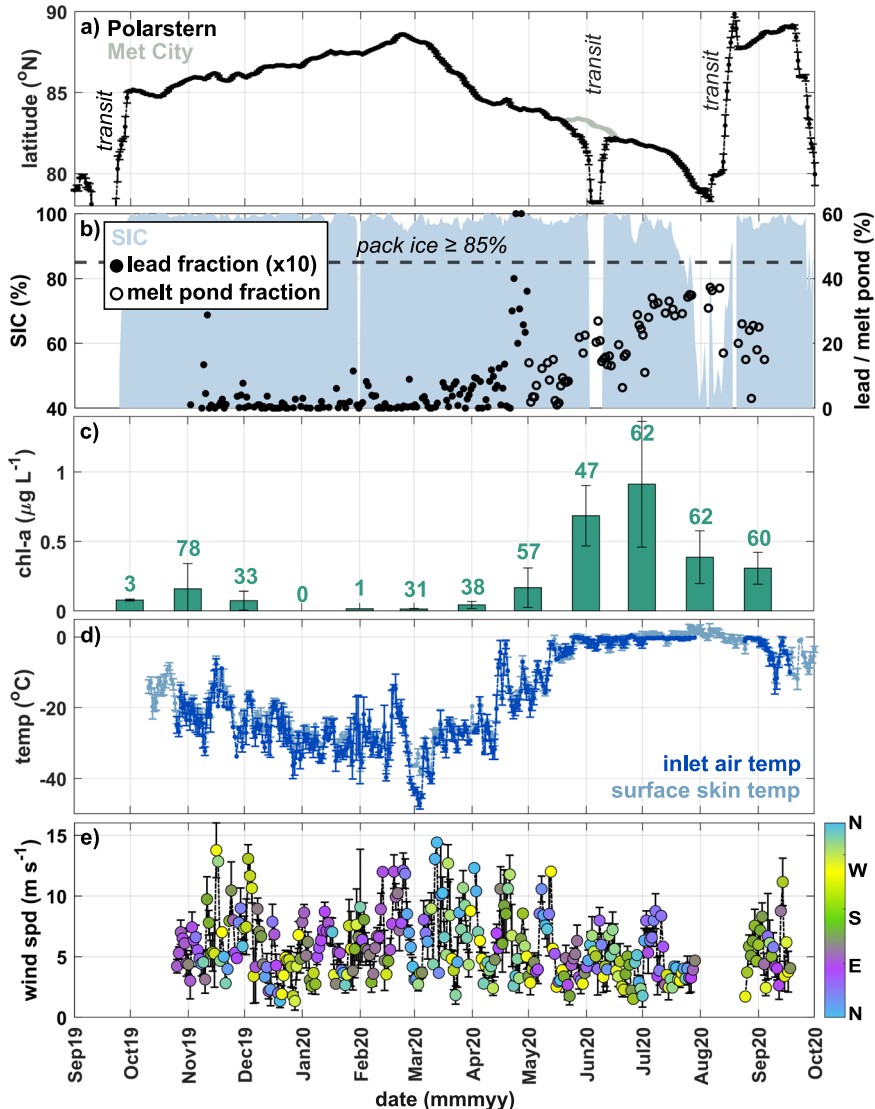

**Fig. 2 Time series of floe location, sea ice, meteorology, and ecosystem data. a** Latitude measured on Polarstern and the Central Observatory (CO) at "Met City", with transit periods denoted. **b** Sea ice concentration (SIC), and percentage of open water in the form of leads and melt ponds within a 1 × 1 degree grid around Polarstern. Lead percentage is multiplied by 10 to show variability. **c** Underway chlorophyll-a (chl-a) concentration measured from samples collected on Polarstern (values = number of discrete samples). **d** Temperature (temp) from the aerosol inlet on Polarstern and at the ice surface, and **e** wind speed (wind spd) colored by direction (N north, E east, S south, and W west), from "Met City". All data aside from chl-a are either collected as or averaged daily. Error bars indicate standard deviation of the calculated averages.

time), especially when airmasses arrived from over the pack ice. Overall, these sea ice, ocean, and meteorological observations drive the annual variability of INPs and shed light on the probable sources of these influential particles.

**The central Arctic INP annual cycle**. INP concentration was lowest in winter and highest in central Arctic summer (Fig. 4), which agrees with year-long Arctic measurements made at lower-latitude, coastal locations[25]. The seasonal cycle was shaped by a combination of a relatively consistent presence of cold-temperature INPs (active at $<-15\,°C$) throughout most of the year with a preponderance of warm-temperature INPs (active at $\geq-15\,°C$) during summer. Cold-temperature INPs (specifically at $-20.0$ and $-22.5\,°C$) peaked during the winter and late summer. The winter peak was likely influenced by dust from continental sources[26–28], as shown by the intermittent transport of continental airmasses from Siberia, Eastern Europe, and Northern

Canada (Supplementary Figs. 4–6 and 11–13) and the presence of mostly inorganic (i.e., mineral) INPs at temperatures below $-20\,°C$ (Supplementary Fig. 16). The late summer peak was likely also from lower latitude sources, as the Polarstern reached the ice-free ocean during transit (Figs. 2a and 3a). Although, these long-range source regions were generally not as influential at the surface compared to more regional sources (Supplementary Figs. 1 and 7).

Warm-temperature INPs, which were predominantly protei-naceous (Supplementary Fig. 16), were most abundant in June and July, with concentrations up to $2\times10^{-2}\,l^{-1}$ at $-15\,°C$, and onset freezing temperatures up to $-6\,°C$ (Fig. 4). This summer INP enhancement coincided with elevated chl-a concentration as well as increased open water and melt pond fractions within the pack ice (SIC and melt pond fraction surrounding Polarstern was 92% and 24% on average, respectively). Local wind speed averaged $4.3\,m\,s^{-1}$ in June and July, while 3-day airmasses predominantly traveled from over the pack ice (73% of the time)

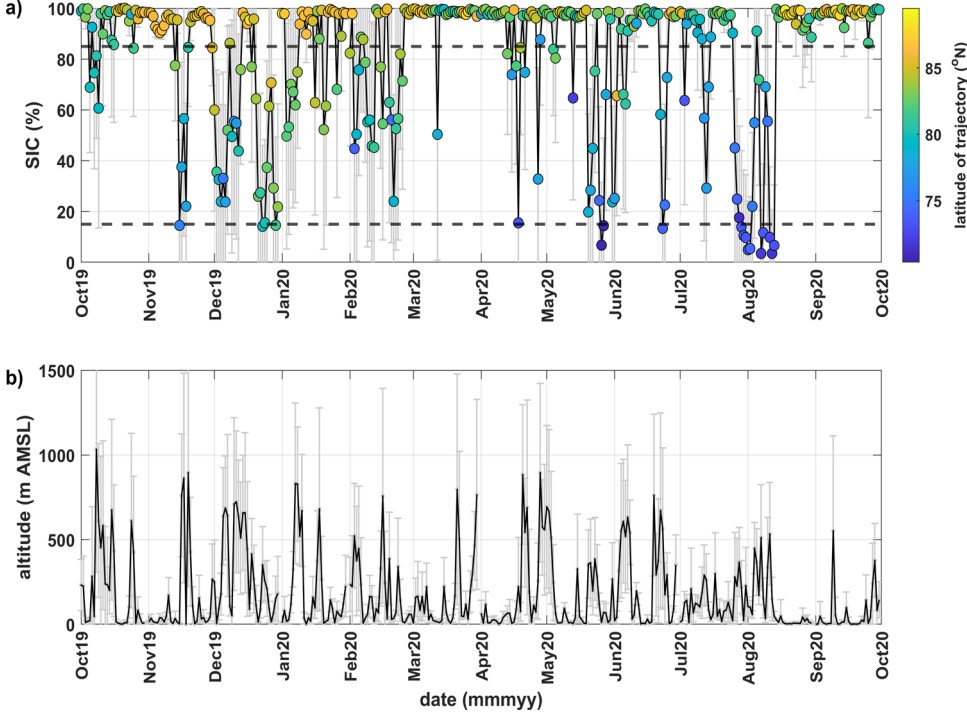

**Fig. 3 Spatial properties of airmass transport pathways to Polarstern. a** The average sea ice concentration (SIC) along each 3-day airmass backward trajectory endpoint. Trajectories were initiated daily at 00:00 UTC and at 30 m above mean sea level (AMSL) above Polarstern with endpoints every 6 h back in time. The dashed lines indicate the thresholds for pack ice (85–100%), the marginal ice zone (15–85%), and ice-free ocean or land (0–15%). Threshold definitions were obtained from the Norwegian Polar Institute (https://www.npolar.no/en/themes/the-marginal-ice-zone/). **b** Corresponding altitude along each 3-day trajectory averaged at all end points. Error bars indicated standard deviation of the daily averages.

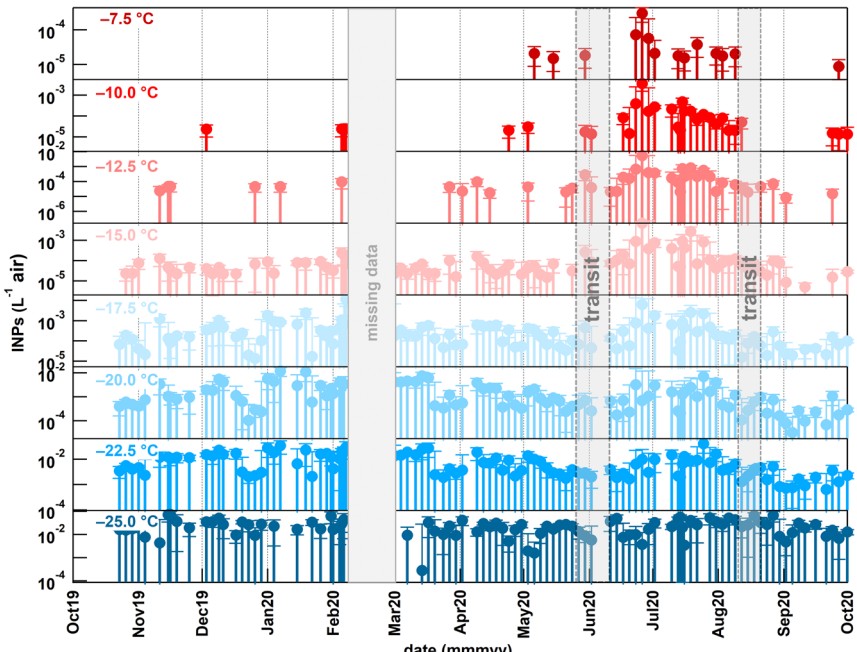

**Fig. 4 Time series of ice nucleating particles (INPs) during the Multidisciplinary drifting Observatory for the Study of Arctic Climate (MOSAiC) year.** INP concentration in per liter of air (l⁻¹ air) at every 2.5 °C interval from –7.5 to –25.0 °C for the total aerosol samples (~150 nm–12 μm; the sum of the INP concentrations from the four size ranges or sample sets collected every 24 h). A subset of the daily sample sets was processed (i.e., ~2 sample sets per week). The gray shaded regions indicate missing data (i.e., where samples were not collected) or transit times when Polarstern traveled between the ice floe and land during the expedition. Note the difference in INP concentration scales between the different temperatures. Error bars represent standard deviation.

and MIZ (20% of the time) within the typical well-mixed summer boundary layer (below 400 m (ref. [28])) at an average height of 158 m. It is important to note that transit periods when the ship was in open water south of the pack ice did not correspond to enhanced INP concentration, suggesting that the larger area of ice-free open ocean (SIC < 15%) further south was not a significant source of the warm-temperature INPs in the summer. This is supported by the fact that marine INPs from open ocean sea spray are typically low in concentration, unless the waters are subject to shallow shelf regions or collapsing phytoplankton blooms[17,29–31]. Previous work corroborates the fact that the annual variation of sea ice algal presence, derived from chl-a data, is controlled mainly by environmental conditions such as sunlight, and less by latitude[32]. Additionally, as melt ponds and the MIZ develop, this exposes materials from under-ice and ice-edge algal blooms that manifest during the melt onset and extend into the summer[33,34] and that may become airborne under appropriate wind regimes. As the melt season progresses, these open water sources contain a mixture of ice algal and marine microbial communities that are more densely populated near the surface, while these microorganisms effectively sink in larger areas of open water that has been devoid of ice for longer periods[33], and therefore, farther away from potential exchange with the atmosphere. Together, the evidence presented here suggests that: (1) summer INPs were likely from biological productivity in open water sources either locally within the pack ice, from farther north over the pack ice, or from the MIZ; (2) transported airmasses interacted with these surface sources; and (3) winds were strong enough to produce sea spray from localized open water sources within the pack ice[35,36]. Future publications will focus on aerosol DNA and fluorescence measurements to corroborate the presence of biological aerosol in the summer.

Previous studies report a similar enhancement in summertime Arctic INPs, whereby peak concentration ranged from $2 \times 10^{-3}$ to $6 \times 10^{-1} \, l^{-1}$ at $-15 \,°C$[25–27,37]. Some studies suggest such INPs are linked to microbial emissions from biota in the seawater and surface microlayer[38–42]. However, most of these studies occurred at or near coastal locations that are typically impacted by both marine and terrestrial, often dust-dominated, sources[43], and are thus likely not representative of sources in the central Arctic. During central Arctic INP measurements from Bigg et al. several decades ago, they observed an average concentration of $\sim 1 \times 10^{-2} \, l^{-1}$ (up to $1 \times 10^{-1} \, l^{-1}$) at $-15 \,°C$ in late August through mid-October and attributed the ice-free ocean as the likely source with smaller contributions from open leads in the pack ice[20,21]. Hartmann et al.[26] reported summertime concentrations up to $1 \times 10^{-1} \, l^{-1}$ at $-15 \,°C$ within the MIZ in July and lower concentrations over the pack ice. More recently, Porter et al.[44] reported INP concentrations up to $1 \, l^{-1}$ at $-15 \,°C$ during their Aug/Sep cruise but also observed lower concentrations in the pack ice. In contrast to these three studies, during MOSAiC, INP concentration was highest in the pack ice at 82.1°N in the summer, reaching $2 \times 10^{-2} \, l^{-1}$ at $-15 \,°C$. Possible explanations for this contrast could be variability in time of year, airmass transport pathways, ice thickness, local lead and melt pond areal fraction, and primary productivity peak location and date; however, long-term annual studies at multiple locations within the pack ice, MIZ, and ice-free ocean would be required to statistically identify the key limiting factors that control INP concentration. Overall, the observed concentrations are consistent with what has previously been reported in the Arctic but are unique in that: (1) they demonstrate that INPs can be abundant over pack ice with minimal inferred source from the ice-free ocean relative to the pack ice and MIZ regions and (2) they show relationships between INPs and environmental conditions over a full annual cycle.

**Distinctive size-resolved behavior of central Arctic INPs.** In addition to total concentration, MOSAiC enabled year-round, size-resolved INP measurements over the central Arctic. This valuable information helps to further elucidate the sources and assess which portion of the total aerosol population is most important for cloud ice formation. Existing studies at lower-latitude surface locations concluded that supermicron aerosols are the most abundant INPs, especially at warm temperatures[17,27,45]. However, this relationship between INP concentration and particle size was not observed during MOSAiC (Fig. 5 for INPs active at −20 °C; Supplementary Fig. 14 for full size-resolved INP spectra).

At the beginning of the study, autumn INPs were coarse (≥3 μm), although total aerosol in this size range was not particularly abundant (1–2 in $10^4$ coarse particles were INPs, on average; Supplementary Fig. 15). In Svalbard (78.9°N)[46] and Alaska (71.2°N)[47], coarse sea spray is present in the autumn prior to freeze up, thus, it is plausible that the low INP concentration observed here (84.8–86.2°N) originated from sea spray, episodically generated by high winds over local leads present in the pack ice (Fig. 2) or from slightly lower latitudes within the MIZ (Fig. 3)[16]. In support of this notion, October–November average latitude, SIC, and height along the transport pathways were 83.6°N, 90%, and 179 m, respectively. Based on previous work, it is plausible that these relatively large INPs were intact or fragmented microbial cells from heterotrophic bacteria or diatoms[29,30], or biogenic organics such as phytoplankton cell exudates[42], fatty acids[48], and polysaccharides[30,39]—the presence of biological and organic INPs during the fall corroborates these sources (Supplementary Fig. 16). Further, sea spray INPs are typically supermicron (up to 5 μm[49]) in nature, either existing in that size range (i.e., cells) or internally mixed with sea salt[30,42,50].

The winter transitioned to relatively high concentrations of small INPs (<340 nm) and total aerosol, probably due to preferential gravitational settling of larger aerosols during transport over longer distances[28,46]. This conclusion is supported by the time airmasses spent over ice-free ocean or land south of Polarstern, which was highest in December (Supplementary Figs. 1 and 7), and the presence of inorganic (i.e., likely mineral) INPs (Supplementary Fig. 16). The largest fraction of INPs to total aerosol in this size range was observed in the winter, equating to 2–6 in $10^6$ total particles (Supplementary Fig. 15). These results are atypical in that sub-500-nm aerosols are usually not a dominant size of ice nucleators[51], attesting to the distinctive behavior of central Arctic INPs. However, larger INPs were still present, albeit in lower concentration, which could be attributed to dust or biological INPs that are active at relatively low temperatures compared to summer.

Another shift occurred during summer, whereby the highest INP concentration was observed for ~1–3 μm particles, even though the total aerosol in this size range was relatively low. Total aerosol number concentration was at its annual maximum, which agrees with previous studies at lower latitudes that report high concentration of aerosols formed from biogenic gases[46,47] or primary biological organic aerosol[52]. The highest fraction of INPs to total aerosol in the ~1–3 μm size range was 3–7 in $10^4$ total particles (Supplementary Fig. 15), indicating more abundant INPs compared to the rest of the year, and especially in the late winter/spring when ~1–3 μm total particle concentration was highest. This transition suggests a change in source of these supermicron-sized particles. The fact that the largest particles were not the most abundant INPs in the summer is somewhat puzzling because ice nucleation is typically observed to be more effective for larger particles[17,27,45]. One possible explanation is preferential emission of particles in this intermediate size range from the local melt ponds and leads. Previous studies report sea spray emission

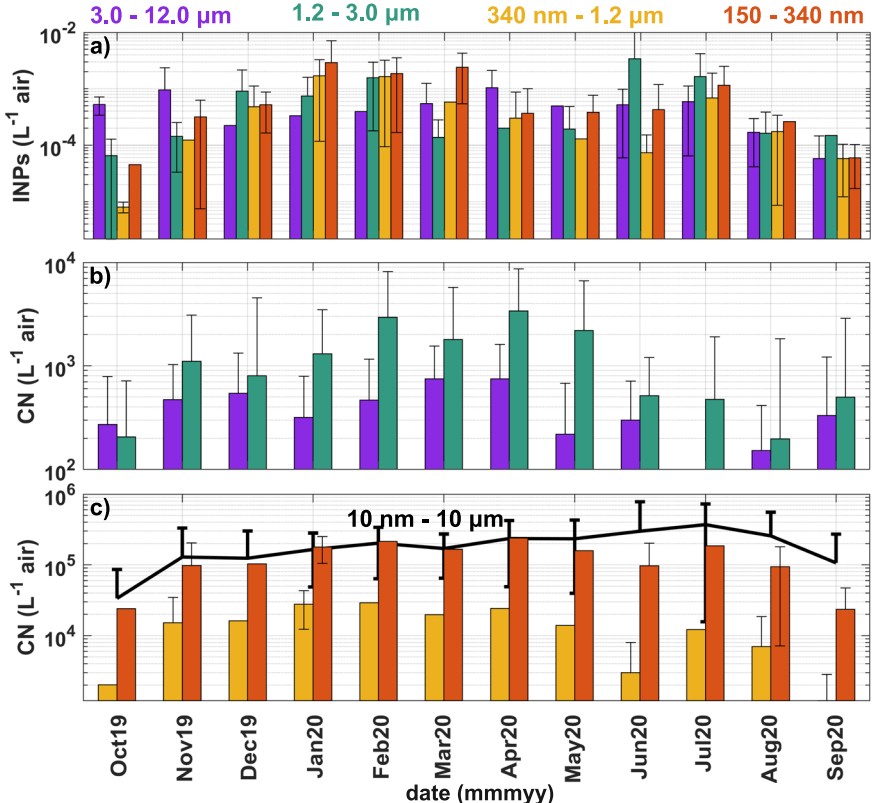

**Fig. 5 Seasonal cycle of size-resolved total aerosol and ice nucleating particles (INPs) active at −20 °C. a** Average cumulative INP concentration per month for particles in the size ranges of 3.0–12 µm, 1.2–3.0 µm, 340 nm–1.2 µm, and 150–340 nm. Monthly averaged total aerosol number concentrations (CN) for **b** coarse aerosols (3.0–12 µm and 1.2–3.0 µm) and **c** smaller fractions (340 nm–1.2 µm and 150–340 nm). Also shown in **c** is the total aerosol number concentration from 10 nm to 10 µm averaged per month. INPs are shown at −20 °C as opposed to warmer temperatures since INPs were active starting at this freezing temperature for most of the year thus to enable an intra-monthly comparison possible (see Fig. 4). Error bars represent standard deviation.

of supermicron particles with a mode around 2 µm from leads at higher wind speeds[35,36]; to our knowledge, no studies reporting size-resolved aerosol from melt ponds exist. This mechanism is plausible as the wind speed was frequently above the minimum threshold for supermicron aerosol generation and both melt ponds and leads were present in June and July (Supplementary Fig. 17). Particles of this size have typical residence times of a few days or less in the atmosphere[53]. Since transport was primarily from over the pack ice at relatively low height (≤500 m; Supplementary Fig. 17), the enhanced 1–3-µm INPs were most plausibly from the central Arctic open water within the pack ice.

**Implications for central Arctic cloud formation.** New insight into the abundance and sources of INPs throughout the central Arctic year helps to address a fundamental gap in understanding cloud formation, lifetime, and properties—all of which have substantial implications for the energy budget over the pack ice. Figure 6 illustrates the annual cycle of select cloud properties during MOSAiC, which we discuss in the context of our INP observations and offer possible implications based on previous work. This region is cloudy throughout the year[8], meaning there is no shortage of opportunities for aerosol-cloud interactions.

The coldest and highest clouds were present in winter, as observed previously[54]. High fractional occurrence of ice in clouds below 3 km (i.e., low-level clouds) in winter implies that cold-temperature mineral INPs that are relatively small, like those observed here, could serve an important role in cloud ice formation. However, the fact that the surface is predominantly

frozen for most of winter limits the potential local sources of INPs. Additionally, the highly stratified boundary layer, characterized by high variance in sub-cloud equivalent potential temperature, limits the vertical movement of INPs. Given that there are seasonal episodic transport pathways in the lower troposphere from lower latitudes in winter[55], it is most plausible that winter INPs affecting clouds have remote sources. The INPs measured near the surface are likely transported aloft and transported downward by mixing and/or precipitation and sublimation.

Spring and autumn clouds were still relatively cold and somewhat elevated above the surface, but enhanced below-cloud mixing (i.e., less variance in equivalent potential temperature[56]) offers the potential for more interaction between clouds and the near-surface environment where coarse, biological or organic sea spray INPs active at cold temperatures[16] could be produced from more localized leads, melt ponds, or the MIZ. Indeed, high ice fractions in low-level clouds are observed at these times of year. Previous work has indicated that cloud formation in the spring is promoted by strong evaporation from newly-opened water, increasing cloud presence and downward longwave radiative flux, contributing to even more sea ice melt[57].

Summer, unlike the rest of the year, had much lower and warmer clouds that did not contain as much ice. For these clouds, only warm-temperature, biological INPs would be capable of forming ice at heights above the melting level. Even so, ice occurred 42–68% of the time in clouds that were typically linked through vertical mixing with the surface. As a result, local biological sources of INPs active at the warmest temperatures

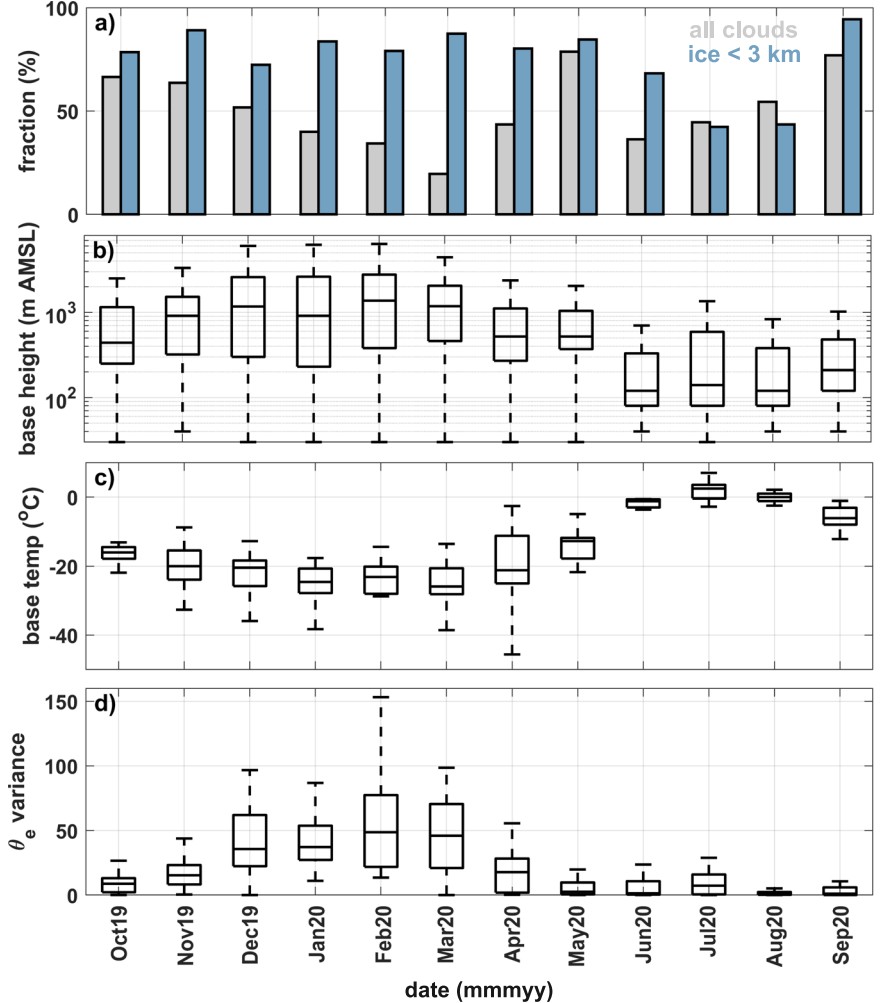

**Fig. 6 Annual cycle of clouds and atmospheric stratification during the Multidisciplinary drifting Observatory for the Study of Arctic Climate (MOSAiC).** Data include monthly averaged: **a** cloud fraction and cloud ice fractional occurrence below 3 km AMSL, **b** cloud-base height, **c** cloud-base temperature (temp), and **d** vertical variance of sub-cloud equivalent potential temperature ($\theta_e$). For **b–d**, the central mark indicates the median, and the bottom and top edges of the box indicate the 25th and 75th percentiles, respectively. The whiskers extend to the most extreme data points not considered outliers.

could have readily been transported vertically to support heterogeneous cloud ice formation aloft. This conclusion is further supported by ref. [58], who found a higher frequency of AMPCs that contained ice at temperatures above −10 °C when the boundary layer was mixed up to cloud level vs. when it was not. Generally, evidence here reveals a seasonal dichotomy of INP sources in the central Arctic that could influence cloud ice processes.

Due to a lack of observations in the central Arctic relative to other regions, models are typically missing a local source of INPs, specifically those from marine biogenic emissions in the summer[42,59]. Recent Earth system model efforts reveal that marine INPs may be important in primary ice formation in Arctic clouds, even more so than dust[60], More broadly, model spread in estimates of Arctic amplification have even been attributed to their sensitivities to INPs, and specifically summertime INP concentrations[61]. However, more observations are needed to further reduce modeling uncertainties associated with INPs and their impacts on Arctic clouds. Specifically, vertical and spatial measurements of INPs in the Arctic are crucial to determine their sources, transport mechanisms, and direct interactions with clouds. As aerosol pollutant emissions decrease[3], amplified warming leads to sea ice retreat[62] and enhanced primary

productivity[63], and the Arctic becomes potentially rainier[64] and cloudier[65]. Understanding the sources of natural aerosols that seed clouds from within the Arctic then becomes essential[3]. Our unique new results raise the possibility that strong seasonal variability in local marine, and episodic long-range transported terrestrial emissions largely control the central Arctic INP population and its subsequent influence on cloud formation and phase. While the presence of cloud liquid water significantly increases cloud radiative effects on the surface, converting some of that liquid to ice reduces the radiative impact[5]. Ultimately, INP-modulated changes in cloud phase partitioning can significantly impact the energy budget at the sea ice surface.

## Methods

**MOSAiC expedition platforms and location**. The MOSAiC international drift expedition took place in the central Arctic pack ice, leaving Tromsø, Norway on 20 September 2019 and returning to Bremerhaven, Germany on 12 October 2020. The expedition was broken down into legs, some of which required transit of Polarstern back to land for exchange of personnel, fuel, and provisions (transits are noted in figures in the main text). Dates for legs 1, 2, 3, 4, and 5 are 20 Sep–13 Dec 2019, 13 Dec 2019–24 Feb 2020, 24 Feb–04 Jun 2020, 04 Jun–12 Aug 2020, and 12 Aug–12 Oct 2020, respectively[22,66]. Measurements presented here were obtained from the MOSAiC Central Observatory (CO), including onboard the German research icebreaker Polarstern and from Met City, which was an installation on the ice in

the CO between 300–600 m from Polarstern[22]. GPS devices operated on Polarstern and at Met City measured latitude and longitude at hourly and 1-min resolution, respectively. Arctic Mapping Tools[67] for MATLAB were used for plotting the maps for the positioning data (and airmass backward trajectories over monthly SIC; see below).

**Surface meteorological data**. Air temperature presented here was measured at a 1-s resolution using a Vaisala WXT520 sensor at the aerosol inlet on the Aerosol Observing System (AOS)[68] from the U.S. Department of Energy's (DOE) Atmospheric Radiation Measurement (ARM) Program, which was positioned on the bow of Polarstern, ~18 m above the sea ice surface[69]. Air temperature was also measured at 2 m above the sea ice using a Vaisala PTU307 sensor package situated on a 10-m meteorological tower at Met City[70] (not shown but used for as a cross check for the temperature measured on Polarstern). Surface skin temperature was measured at a 1-min resolution using an Apogee SI-4H1 infrared thermometer mounted at 2 m on the Met City tower and pointed down at the surface. Winds were measured at a 1-min resolution using a sonic anemometer mounted at a nominal height of 10-m on the Met City tower[22]. For the May–June period when Met City was not installed, these measurements were obtained from an identical sonic anemometer mounted at a nominal height of 3 m on an Atmospheric Surface Flux Station that was left in place of the Met City installation[71]. $U$ (eastward) and $V$ (northward) wind vectors were calculated to obtain daily averages using Eqs. (1) and (2), respectively:

$$U = -1 \times WS \times \sin\left(WD \times \frac{\pi}{180}\right) \qquad (1)$$

$$V = -1 \times WS \times \cos\left(WD \times \frac{\pi}{180}\right) \qquad (2)$$

where WS and WD are wind speed and direction, respectively, in the direction from which winds originate. Winds were also measured on Polarstern but were not used here as they were subject to interference from the vessel. Air temperature, however, was not significantly influenced by Polarstern on a daily-averaged resolution.

**Atmospheric structure**. Radiosondes measuring profiles of pressure, temperature, humidity, and winds were launched every 6 h from the helicopter deck on the stern of Polarstern up to ~20 km[72]. Equivalent potential temperature ($\theta_e$) was calculated following Eqs. (22) and (43) in ref. [73] for the radiosonde profiles. From that, daily mean vertical $\theta_e$ variance between the launch platform and the daily mean cloud base height as retrieved from ceilometer (see details below) were employed as a proxy for sub-cloud mixing. Likewise, daily mean temperature at the estimated cloud base was extracted from the soundings.

**Cloud macrophysical and microphysical properties**. Cloud fraction and base height were estimated using 15-s data from a ceilometer (Vaisala CL31) operated on the upper deck of Polarstern as a part of the second ARM Mobile Facility (AMF2)[74–76]. The first or lowest cloud base height (out of three possible for multilayer cloud scenarios) was used as that is the most relevant for interactions with aerosols detected near the surface. Cloud fraction over time was calculated using the first cloud base height data from the ceilometer. Each 15-s interval was classified as 0 (no cloud base detected) or 1 (first cloud base detected) and monthly mean cloud fractions were calculated from there. Cloud ice water path (IWP) at a 1-min resolution was retrieved via a multi-sensor cloud property retrieval method[77]. The fractional occurrence of cloud ice was calculated based on when IWP > 0 for data below 3 km, then taking the total minutes flagged for ice divided by the total minutes where clouds were detected below 3 km per day.

**Airmass trajectory modeling**. Airmass backward trajectories were calculated using HYSPLIT, the HYbrid Single Particle Lagrangian Integrated Trajectory model with the SplitR package for RStudio (https://github.com/rich-iannone/SplitR)[78,79]. Reanalysis data from the National Centers for Environmental Prediction were used as the meteorological fields in HYSPLIT simulations (2.5-degree latitude-longitude; 6-h). Trajectories were initiated from 30 m AMSL and from the latitude and longitude of Polarstern at 00:00 daily. The initiation height of 12 m above the AOS inlet was used to estimate larger-scale airmass movements outside of the possible wind shear interference from Polarstern's decks. An initiation height of 100 m AMSL was also used to demonstrate that possible disturbances from the underlying surface did not change the main conclusions and to validate the results from the trajectories ending at 30 m. Trajectories were calculated 3 days back in time to account for average atmospheric aerosol residence times for supermicron aerosol (i.e., typically residence times of a few days or less in the atmosphere)[53], as these aerosol sizes were the most proficient INPs observed during MOSAiC. Additionally, 5- and 7-day trajectories were calculated to account for possible longer residence times of smaller particles. Recent work utilized 7-day trajectories for coarse mode aerosol source apportionment in Svalbard[46]. It is important to note that the determination of the length of back trajectories is a compromise between typical aerosol lifetimes and the uncertainty in calculation that increases with time. Additionally, particle sphericity impacts the rate of gravitational settling[80] but neither particle sphericity or gravitational settling are accounted for

here, as the relevant measurements to determine particle sphericity were not made and HYSPLIT is a passive air parcel model. Figure 3 and Supplementary Figs. 2 and 3 show these trajectories averaged over each of the 3, 5, or 7 days for 30 m AMSL and Supplementary Figs. 8–10, which includes average latitude and altitude and SIC within a 0.1-degree grid box at each 6-h endpoint along the trajectories. A comparison of trajectories for 3, 5, and 7 days back in time is shown in Supplementary Figs. 1 and 7 for 30 m and 100 m AMSL, respectively, demonstrating the similarity between the various parameters over the calculated lengths.

**Sea ice, melt pond, open lead, and oceanographic data**. Daily SIC data (https://nsidc.org/data/g10005, Version 1) were obtained from the National Snow and Ice Data Center[81] and were derived from the Multi-sensor Analyzed Sea Ice Extent Advanced Microwave Scanning Radiometer 2 MASIE-AMSR2 (MASAM2) daily 4-km SIC product that is a blend of two other daily sea ice data products: ice coverage from the product at a 4-km grid cell size and ice concentration from the AMSR2 at a 10-km grid cell size[82]. MASAM2 was used to meet a need for greater accuracy and higher resolution in ice concentration fields. Daily SIC averaged within a 1-degree grid box around Polarstern and within a 0.1-degree grid box around airmass backward trajectory endpoints were computed and shown in Figs. 2 and 3 and Supplementary Figs. 2 and 3. Monthly Arctic-wide SIC averages were calculated and shown in Fig. 1 and Supplementary Figs. 4–6 and 11–13. Conventional definitions from the Norwegian Polar Institute for the pack ice (85–100%), MIZ (15–85%), and open water or land (0–15%) were used.

Lead fractions at 12.5 km² spatial resolution were derived from a daily 1 km² binary lead product that is based on Moderate Resolution Imaging Spectroradiometer (MODIS) thermal infrared satellite imagery[83]. In this retrieval, a lead is attributed to a significant positive surface temperature anomaly in the cold winter sea-ice surface. The lead fraction thus represents a spatially integrated measure of the lead activity around the CO. Lead fraction product data are only available for November through April. Melt pond fractions at 12.5 km² spatial resolution were derived from a daily melt pond fraction product based on the inversion of a forward model applied to cloud-screened optical measurements made by the Ocean and Land Colour Instrument aboard the Sentinel-3 satellite[84]. The melt pond fraction represents the areal fraction of melt ponds on top of the sea ice portion. Daily lead and melt pond fractions were averaged within a 1-degree grid box around Polarstern. Melt pond fraction product data are only available for May through August.

Samples for the determination of surface chlorophyll (chl-a) were taken from 11-m water depth via Polarstern's underway system. Depending on chl-a levels, 2–4 l of seawater were filtered onto glass microfiber filters (Whatman®, Grade GF/F) in duplicates or triplicates and frozen at −80 °C until further analyses. Samples were extracted in 90% acetone over night at 4 °C and subsequently analyzed on a fluorometer (TD-700; Turner Designs, USA), including an acidification step (1 M HCl) to determine phaeopigments following ref. [85]. Most samples were analyzed at the Alfred Wegener Institute (AWI) after the campaign had ended. During leg 3, one subsample per sampling event was measured on board within 3 days after sampling, while a second subsample was analyzed at AWI (5–9 months after sampling). No systematic difference between the replicates could be detected, indicating no significant degradation of chl-a took place during storage and transport.

**Aerosol number concentrations and size distributions**. Aerosol number concentrations from 10 nm to 10 μm at a 1-s resolution were measured using a condensation particle counter (CPC model 3772; TSI, Inc.) located in the AOS[86,87]. Submicron aerosol size distributions from 60 nm to 1 μm were measured at a 10-s resolution using an ultra-high sensitivity aerosol spectrometer (UHSAS; Droplet Measurement Technologies, Inc.) also located in the AOS[88,89]. The UHSAS measurements were converted to aerodynamic diameter ($D_a$), assuming optical diameter is equivalent to the true diameter ($D_{true}$), using the following equation and density of 1.5 g cm$^{-3}$ based on estimates from previous studies based in pristine and polluted locations[90,91]:

$$D_a = D_{true} \times \sqrt{1.5} = 1.22 \times D_{true}$$

Additional aerosol size distributions that cover the supermicron/coarse size range were measured at a 20-s resolution from 500 nm to 20 μm using an aerodynamic particle sizer (APS model 3321; TSI, Inc.) that was operated by École Polytechnique Fédérale de Lausanne in the "Swiss container", adjacent to the AOS. The AOS and Swiss container had separate but similar inlet stacks to maximize transmission efficiency of up to 20-μm and 40-μm particles, respectively, through protective components (e.g., precipitation guards) and sufficiently high flow rates[92–94]. The whole air inlet of the Swiss container is built after the Global Atmosphere Watch recommendations[95]. The AOS inlet contained a back-pressure purge system that was applied to minimize sampling of ship emissions, such that during obvious periods of ship emission influences, the purge system would cause the inlet to stop sampling while all instruments remained operational[96]. However, not all periods were detected and purged. Thus, developing and applying a pollution mask to omit aerosol data affected by local contamination (e.g., ship stacks and vents, snow machines) was essential. The basic principle of the pollution detection algorithm is based on the time derivative of the particle number concentration. Local pollution was characterized by strong fluctuations (and thus high time derivatives) in particle

number concentration. If this time derivative exceeds a certain threshold, the data were flagged as polluted. The basic principle of this method was developed and used for the 2018 Microbiology-Ocean-Cloud-Coupling in the High Arctic campaign on the Swedish ice breaker Oden[97].

**Size-resolved aerosol collection and offline INP analysis**. Size-resolved aerosols were collected from 23 October 2019 to 1 October 2020 using the Colorado State University (CSU) 4-stage Davis Rotating-drum Unit for Monitoring cascading impactor (DRUM model DA-400; DRUMAir™)[17,27,98,99] through the AOS inlet, in parallel with the CPC and UHSAS. The DRUM collected daily integrated samples at 29–34 l min$^{-1}$ (average flow rate of 31.6 ± 1.6 l min$^{-1}$) from 0.15 to >12 μm in diameter (size cuts at 2.96, 1.21, and 0.34 μm) on sterilized perfluoroalkoxy substrate strips coated with petrolatum—a material containing very few artifacts that interfere with INP analysis. It is important to note that the AOS inlet has a high transmission efficiency for particles from 10 nm to 4 μm but has large uncertainties in transmission efficiency above 4 μm due to low ambient aerosol signal in that size range, thus it is possible the DRUM samples produced lower INP concentration estimates in that size range as well[92]. Sample start times ranged from 10:25–15:15 UTC (average time of 14:30 UTC) and collected continuously for 24 h. Aerosols were "smeared" onto the sample strips as the discs in each stage rotated slowly over time (5 mm of aerosol loading) followed by quick 2 mm rotation, affording a sample blank in between each daily sample. Weekly flow and pressure checks were conducted throughout the year to ensure obstructions were not present in the orifices. Samples were preserved and shipped frozen (≤–20 °C) until analysis at CSU.

The CSU cold plate is an established method used to measure immersion mode INPs[17,27,43,99,100]. For preparation, aerosol samples were added to 2 ml of 0.1-μm filtered deionized water and mixed for at least 20 min at 200–500 rpm in a laminar flow clean hood to resuspend particles into a suspension. After preparation, 100 × 2.5-μl aliquots were created on a 3-inch diameter copper plate and covered to prevent contamination. The plate was cooled at ~1–10 °C min$^{-1}$ from room temperature until all drops froze on the plate or until the cold plate limit of ~ –29 °C. Drop freezing was detected and recorded through monitoring software to provide the freezing temperature and cooling rate for each drop. Each sample was tested three times (i.e., three sets of 100 drops).

For MOSAiC, a subset of the 1280 DRUM samples was processed and analyzed using the cold plate. Approximately every third day's sample set was chosen—a "sample set" corresponds to all four samples from each of the size ranges, per 24-h period, equating to analysis of 388 samples. However, this time resolution varied as samples that were not under the AOS inlet purge were selected. The purging reduced possible contamination from Polarstern, as recent work has shown that ship pollution potentially introduces INPs at low temperatures (i.e., < –22 °C)[101]. However, this recent work analyzed deposition-mode INPs, which likely do not overlap with the properties of immersion-mode INPs, as other previous studies have demonstrated how pollution aerosol, in general, is not a sufficient source of immersion INPs[102–106].

**Total aerosol collection and offline INP treatments**. Total aerosol samples were collected on top of the AOS container near the inlet from 23 October 2019 to 1 October 2020 using single-use Nalgene™ Sterile Analytical Filter Units that were prepared by replacing their cellulose nitrate filters with 0.2-μm polycarbonate filters backed with 10-μm polycarbonate filters (each are 47 mm diameter Whatman® Nuclepore™ Track-Etched Membranes), both pre-cleaned at CSU[107]. The filter units were open-faced, secured outside to the AOS railing and shielded from precipitation and blowing snow. Vacuum line tubing connected the filter unit to the flow meter followed by vacuum pump, both of which were housed inside the AOS container. Average flow rate was 21.9 l min$^{-1}$. Samples were started at 14:15 UTC on average and collected continuously for 72 h. Following collection, filters were removed from the single-use units, placed in sterile Petri dishes, and preserved and shipped frozen (≤ −20 °C) until analysis at CSU.

Samples were processed using the CSU Ice Spectrometer (IS)[29,31,108–113], a technique which is comparable to the cold plate method[114]. Filters were placed in sterile centrifuge tubes, 8 ml of 0.1 μm-filtered deionized water added, and particles re-suspended by tumbling end-over-end on a rotator for 20 min. From these suspensions, 32 50-μl aliquots were distributed into sterile, 96-well PCR trays in a laminar flow cabinet. Serial dilutions were applied to cover INP concentrations over the full temperature range. Plates were then placed on the blocks of the IS, the device covered with a plexiglass window, and the headspace purged with filtered N$_2$. The device was cooled at 0.33 °C min$^{-1}$ until ~ –29 °C using a recirculating low temperature bath, and the freezing of wells recorded every 0.5 °C via a CCD camera system. Thermal treatments and peroxide digestions provide valuable insights into INP composition. Heat treatments were performed on approximately one third of the IS filter samples by heating 2 ml of suspension to 95 °C for 20 min to denature heat-labile INPs, such as proteins[110,112,115]. Hydrogen peroxide (H$_2$O$_2$) digestions were performed on a further 2 ml of suspension to remove all bio-organic material. This procedure, and the neutralization of remnant H$_2$O$_2$ to prevent freezing point depression[112], the samples were then re-analyzed to assess the reduction caused by the decomposition of heat-labile and all organic INPs, respectively.

**Calculating cumulative INP concentration**. For both techniques, the fraction of drops frozen at each temperature interval and the known total volume of air sampled were used to calculate INP concentration with a universally-used equation[116]:

$$K(\theta)(L^{-1}) = \frac{\ln(1-f)}{V_{\text{drop}}} \times \frac{V_{\text{suspension}}}{V_{\text{air}}}$$

where $f$ is the proportion of droplets frozen, $V_{\text{drop}}$ is the volume of each drop, $V_{\text{suspension}}$ is the volume of the suspension, and $V_{\text{air}}$ is the volume of air per sample. See Supplementary Fig. 14 for all size-resolved cumulative INP spectra and Supplementary Fig. 15 for the ratio of each size-resolved INP concentration per month at −20 °C to total aerosol (complimentary to Fig. 5 in the main text). For the treatment spectra, the difference in the INP-temperature spectra after each treatment determines the influence of that INP type in the original sample, and the residual spectrum gives the assumed mineral INP component. Thus, this processing provides four key measures from each sample: total, heat-labile (i.e., biological), bio-organic, and inorganic (i.e., probably mineral dust) INP concentrations. However, the inorganic INP fraction could also contain minor contributions from exotic mineral marine INPs[117]. Treatment results for select temperatures throughout the MOSAiC expedition are shown in Supplementary Fig. 16. For both sample collections, certain periods have missing data as samples were not collected during those periods or sample strips/filters were contaminated. The total and size-resolved INP data and metadata are publicly-available on the DOE ARM Data Archive[118,119].

## Data availability

Met City meteorological data are available from the National Science Foundation's Arctic Data Center (https://arcticdata.io/catalog; https://doi.org/10.18739/A2VM42Z5F and https://doi.org/10.18739/A2CJ87M7G). AOS meteorological data on Polarstern are available on the DOE ARM Data Archive (https://www.arm.gov/data/; https://doi.org/10.5439/1025153). Radiosonde and Polarstern positional data are available from PANGAEA (https://www.pangaea.de/; https://doi.org/10.1594/PANGAEA.928656). Sea ice data (Version 1) is available from the National Snow and Ice Data Center (https://nsidc.org/data/g10005; https://doi.org/10.7265/N5ZS2TFT). The INP (https://doi.org/10.5439/1804484 and https://doi.org/10.5439/1798162 for total and size-resolved INPs, respectively), CPC (https://doi.org/10.5439/1046184), UHSAS (https://doi.org/10.5439/1409033), and cloud parameter datasets (https://doi.org/10.5439/1181954) are available on the DOE ARM Data Archive (https://www.arm.gov/data/). Melt pond data are available from The University of Bremen (https://seaice.uni-bremen.de/melt-ponds/). Metorological field data used in HYSPLIT simulations are availble from the National Oceanic and Atmospheric Administration (https://www.ready.noaa.gov/archives.php). The APS, cholophyll, and lead fraction datasets generated during and/or analyzed during the current study are available from the corresponding author on reasonable request. The final versions of these data will be made publically available in the near future.

## Code availability

SplitR package for RStudio used to calculate HYSPLIT back trajecotries are available from GitHub (https://github.com/rich-iannone/SplitR). Arctic Mapping Tools for MATLAB used for plotting the maps for the positioning data are available from MathWorks File Exchange (https://www.mathworks.com/).

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

## Acknowledgements

This work was carried out, and data used in this manuscript were produced, as part of the international Multidisciplinary drifting Observatory for the Study of Arctic Climate (MOSAiC20192020). The authors would like to thank all persons involved in the expedition of the Research Vessel Polarstern during MOSAiC in 2019–2020 (AWI_PS122_00)[120]. A subset of the data was obtained from the Atmospheric Radiation Measurement (ARM) User Facility, a U.S. Department of Energy (DOE) Office of Science User Facility Managed by the Biological and Environmental Research Program. Radiosonde data were obtained through a

partnership between the leading Alfred Wegener Institute (AWI), the Atmospheric Radiation Measurement (ARM) User Facility, and the German Weather Service (DWD). L. Wischnewski, T. Brenneis and A. Terbrüggen are acknowledged for their help with chl-a measurements. This work was funded by the DOE ARM and Atmospheric System Research (ASR) programs (DE-AC05-76RL01830, DE-2204 SC0019745, DE-SC0019251, DE-SC0021341) for J.M.C., K.B., T.C.J.H., C.H., P.J.D. and M.D.S.; the U.S. National Science Foundation (NSF) Office of Polar Programs (OPP-1724551) for E.C. and J.B.; and the German Federal Ministry for Education and Research (BMBF) for C.J.M.H. and A.F. through financing the AWI Helmholtz Zentrum für Polar und Meeresforschung and the Polarstern expedition (N-2014-H-060_Dethloff). I.B. received funding from the Swiss National Science Foundation (grant no. 200021_188478). J.S. received funding from the Swiss Polar Institute and holds the Ingvar Kamprad Chair for Extreme Environments Research sponsored by Ferring Pharmaceuticals.

## Author contributions

J.M.C. participated in the MOSAiC expedion, was the lead of INP sample collection, obtained and prepared data for the manuscript, and was the lead author. K.B., T.C.J.H., C.H., and P.D. were responsible for INP sample processing. M.D.S. participated in the expedition and provided guidance on Met City and cloud radar data. S.D. participated in the expedition and provided radiosonde and calculated variance data. S.W. provided the lead fraction data. J.S. and I.B. participated in the expedition and provided APS data. C.J.M.H., A.F., E.C., and J.B. participated in the expedition and were responsible for the chl-a data. R.S. provided guidance on the melt pond areal fraction data. O.P. provided insight into larger scale transport. All authors contributed to the writing of the manuscript.

## Competing interests

The authors declare no competing interests.
