## [Peer Review File · Nature Communications]

First annual cycle observations of aerosols capable of ice formation in central Arctic cloudsREVIEWER COMMENTS

Reviewer #1 (Remarks to the Author):

Comments on “First annual cycle observations of aerosols that seed ice formation in central Arctic clouds” by Creamean et al.

The manuscript by Creamean et al. presents the seasonal variations in the concentration, size distribution and possible sources of ice nucleating particles (INPs) in the central Arctic based on a year-round cruise observation. The results are useful to provide cloud parameters in climate models. Whereas, in my opinion, some explanations on the results are not so sound. The influencing factors and the sources of INPs are mainly interpreted based on environmental conditions and air mass trajectories, but direct evidences (e.g., chemical tracers, microbial abundance and compositions) are lack. Direct measurements of aerosol/INP chemistry are required to more accurately identify the sources and influencing factors of INP populations in the central Arctic.

My specific comments are listed as follows:

1. Title: “aerosols that seed ice formation in central Arctic clouds”. The results of the presented study show the potential of the aerosol particles in the central Arctic that may participate in ice cloud formation rather than the aerosol particles seed ice formation in central Arctic clouds. The title should be revised to be more accurate.

2. Line 52-54: Previous studies have reported that ash, soot or organic particles may also be ice nucleation-active, although the results are still controversial (Cozic et al., 2007, 2008; Levin et al., 2016; Umo et al., 2015; McCluskey et al., 2017; Wolf et al., 2020; Wilson et al., 2015). How about the contributions of other particle types (e.g., ash, soot or organics) to INPs in the Arctic, in addition to mineral and biological materials? Moreover, understanding the compositions of the INPs is crucial to capture the breadth of INPs (Line 59), although some studies might suggest that anthropogenic pollution does not contribute to atmospheric INPs.

3. Line 84-86: The sum percentages of air masses over the pack ice, MIZ and open water or land are not 100%. What does “north of 80°N (79% of the time) mean”? In addition, the authors set the initial height of backward trajectories as 12 m above the AOS (Line 63 in the supplementary), which might ineffectually estimate the trajectories of air masses because the meteorological conditions of the very-near surface of the ocean may be inaccurate due to intense turbulences.

4. Line 94-99: Are the INP concentrations illustrated in each panel of Fig. 5 accumulated for every 2.5°C interval, or newly frozen ones in each 2.5°C interval? The results described here are not obviously illustrated in Fig. 5. Cold-temperature INPs during the winter were likely comparable to those during other seasons.

5. Line 102-110: The two results reported here are somewhat self-contradictory. Abundant warm-temperature INPs coincided with elevated Chl-a concentration as well as increased open water and melt pond fractions within the pack ice, whereas the authors concluded that ice-free ocean was not a significant source of the warm-temperature INPs in the summer based on the unenhanced warm-temperature INPs during the transit periods in open water. The authors inferred summer INPs were likely from biological productivity in open water sources either locally within the pack ice or from farther north over the pack ice. These indications are confusing. How can sea spray particles be produced by winds only from localized open water sources within the pack ice but not ice-free water? What are the possible mechanisms? More direct evidences on the sources of the INPs are required, e.g., chemical and microbial compositions of the Arctic aerosols.

6. Line 116-127: The authors stated that the results that INP concentration was highest in the pack ice at 82.1°N in the summer during MOSAiC contrasted with the results of previous studies that the contribution of INPs from ice-free ocean was significant during summertime. What could be the possible reasons?

7. Line 139-144: In autumn (October, November 2019 and September 2020), a large fraction of INPs are supermicron in size. The authors infer those INPs were sea spray aerosols. What are the efficient INP compositions in the sea spray aerosols? Microbes or biomoleculars (DeMott et al., 2016; McCluskey et al., 2017; Wilson et al., 2015)? Why were those coarse particles not deposited onto the sea surface during long-distance transport? In addition, a minor fraction of INPs are submicron in size. What might they be?

8. Line 145-150: Supermicron particles rather than submicron particles are dominant INPs as reported in previous studies (Córdoba et al., 2021; Line 135-137). Why were the results here different from the results of previous studies? Is it related to the distance or the path the airmasses transported? What are the possible contributors to the small INPs in winter?

9. Sect. “Implications for central Arctic cloud formation”: What could be the relative contributions of locally emitted INPs and long-range transported terrestrial INPs in the central Arctic in different seasons?

10. Line 109-116, 134-137 in the supplementary: Did the size ranges measured by CPC, UHSAS and APS uniformed to aerodynamic size for comparison? The particle size cuts of the DRUM sampler should be determined by the sampling flowrate. There was a large variation in the flowrate (20-35 L min⁻¹). How could this variation influence the size ranges of the particle sampled in each stage? This will largely affect the results illustrated in Figure 5.

References:

Cozic, J., Verheggen, B., Mertes, S., Connolly, P., Bower, K., Petzold, A., Baltensperger, U., and Weingartner, E.: Scavenging of black carbon in mixed phase clouds at the high alpine site Jungfraujoch, *Atmos. Chem. Phys.*, 7, 1797–1807, 2007.

Cozic, J., Mertes, S., Verheggen, B., Cziczo, D. J., Gallavardin, S. J., Walter, S., Baltensperger, U., and Weingartner, E.: Black carbon enrichment in atmospheric ice particle residuals observed in lower tropospheric mixed phase clouds, *J. Geophys. Res.-Atmos.*, 113, D15209, 2008.

Córdoba, F., Ramírez-Romero, C., Cabrera, D., Raga, G.B., Miranda, J., Alvarez-Ospina, H., Rosas, D., Figueroa, B., Kim, J.S., Yakobi-Hancock, J., Amador, T., Gutierrez, W., García, M., Bertram, A.K., Baumgardner, D., Ladino, L.A., 2021. Measurement report: Ice nucleating abilities of biomass burning, African dust, and sea spray aerosol particles over the Yucatán Peninsula. *Atmos. Chem. Phys.* 21, 4453-4470.

DeMott, P.J., Hill, T.C., McCluskey, C.S., Prather, K.A., Collins, D.B., Sullivan, R.C., Ruppel, M.J., Mason, R.H., Irish, V.E., Lee, T., Hwang, C.Y., Rhee, T.S., Snider, J.R., McMeeking, G.R., Dhaniyala, S., Lewis, E.R., Wentzell, J.J., Abbatt, J., Lee, C., Sultana, C.M., Ault, A.P., Axson, J.L., Diaz Martinez, M., Venero, I., Santos-Figueroa, G., Stokes, M.D., Deane, G.B., Mayol-Bracero, O.L., Grassian, V.H., Bertram, T.H., Bertram, A.K., Moffett, B.F., Franc, G.D., 2016. Sea spray aerosol as a unique source of ice nucleating particles. *Proc. Natl. Acad. Sci. USA* 113, 5797-5803.

Levin, E. J. T., McMeeking, G. R., DeMott, P. J., McCluskey, C. S., Carrico, C. M., Nakao, S., Jayarathne, T., Stone, E. A., Stockwell, C. E., Yokelson, R. J., and Kreidenweis, S. M.: Ice nucleating particle emissions from biomass combustion and the potential importance of soot aerosol, *J. Geophys. Res.*, 121, 5888–5903, 2016.

McCluskey, C.S., Hill, T.C.J., Malfatti, F., Sultana, C.M., Lee, C., Santander, M.V., Beall, C.M., Moore, K.A., Cornwell, G.C., Collins, D.B., Prather, K.A., Jayarathne, T., Stone, E.A., Azam, F., Kreidenweis, S.M., DeMott, P.J., 2017. A Dynamic Link between Ice Nucleating Particles Released in Nascent Sea Spray Aerosol and Oceanic Biological Activity during Two Mesocosm Experiments. *Journal of the Atmospheric Sciences* 74, 151-166.

Umo, N. S., Murray, B. J., Baeza-Romero, M. T., Jones, J. M., Lea-Langton, A. R., Malkin, T. L., O'Sullivan, D., Neve, L., Plane, J. M. C., and Williams, A.: Ice nucleation by combustion ash particles at conditions relevant to mixed-phase clouds, *Atmos. Chem. Phys.*, 15, 5195–5210, 2015.

Wilson, T.W., Ladino, L.A., Alpert, P.A., Breckels, M.N., Brooks, I.M., Browse, J., Burrows, S.M., Carslaw, K.S., Huffman, J.A., Judd, C., Kilhau, W.P., Mason, R.H., McFiggans, G., Miller, L.A., Nájera, J.J., Polishchuk, E., Rae, S., Schiller, C.L., Si, M., Temprado, J.V., Whale, T.F., Wong, J.P.S., Wurl, O., Yakobi-Hancock, J.D., Abbatt, J.P.D., Aller, J.Y., Bertram, A.K., Knopf, D.A., Murray, B.J., 2015. A marine biogenic source of atmospheric ice-nucleating particles. *Nature* 525, 234-238.

Wolf, M.J., Zhang, Y., Zawadowicz, M.A., Goodell, M., Froyd, K., Freney, E., Sellegri, K., Rosch, M., Cui, T., Winter, M., Lacher, L., Axisa, D., DeMott, P.J., Levin, E.J.T., Gute, E., Abbatt, J., Koss, A., Kroll, J.H., Surratt, J.D., Cziczo, D.J., 2020. A biogenic secondary organic aerosol source of cirrus ice nucleating particles. *Nat. Commun.* 11, 4834.

Reviewer #2 (Remarks to the Author):

I support publication.

Long term (more than a few weeks) measurements of ice nucleating particles (INPs) are relatively rare. A seasonal cycle in the Arctic is indeed a unique dataset – all the more relevant because of the importance of ice in mixed phase clouds in the Arctic, as the authors note. Further, these measurements are size selected. As rare as documentation of a seasonal cycle of INPs is... Size selected is even more rare.

Is this the first study to report a seasonal cycle of size selected INPs? I don't recall another one. The longest period in Mason et al. (ACP, cited in the paper) is a few months. If this is indeed the first study to show this long a time series of size selected INPs, that should be mentioned more prominently.

The time series of INP as a function of temperature is quite striking. My only quibble with that figure is that the scale on the ordinate axis changes with temperature. This can be quite misleading. For example, one might conclude that the concentrations for -7.5 and -10 are the same until noticing the factor of 10 difference in the scale. Note that it is MUCH MUCH easier to notice such a scale change in the review version of the paper where the figure is scaled up to roughly the size of a page. At least note this change in scale in the caption. (I'm not 25 any more... My eyesight isn't as good as it used to be, and it would be much harder to tease this out in the final version of the article where the figures are much smaller.)

I also have one cautionary note to add concerning the size of the particles. Figure 5 is quite striking, and it is very interesting to me that the smaller particles make up the majority of INP in some months (January, for example). I think the argument based on transport is an interesting one, but some caution is warranted there. The shape of the particles is unknown and nonspherical particles will settle slower than spherical ones of the same volume. See Yang et al for one discussion of that. It's also worth noting that the fraction of particles in a given size range which act as INP isn't shown. Take January for example. The absolute number of particles that act as INP is dominated by the 150 to 340 nm size range but there are also many more of those particles. My back of the envelope, reading rough numbers off the graphs calculation indicates that 1 in a million particles was an INP (at -20) in both size ranges. That's an interesting number, in itself, but also an interesting perspective on the dominance of the smaller size range as INP. (This doesn't account for the fact that the larger particles are very likely to be irregularly shaped, so the aerodynamic size that's reported here is hard to convert to a surface area...)

In summary, the authors have presented quite an interesting dataset, worthy of publication in Nature Communications, in my opinion. I support publication.

Yang, W., Marshak, A., Kostinski, A.B. and Várnai, T., 2013. Shape-induced gravitational sorting of Saharan dust during transatlantic voyage: Evidence from CALIOP lidar depolarization measurements. *Geophysical Research Letters*, 40(12), pp.3281-3286.

Reviewer #3 (Remarks to the Author):

This is the first annual cycle observation of aerosols that seed ice formation in central Arctic clouds. The paper is well-written and reports interesting findings which are important for understanding the regional climate in one of the most sensible regions of the globe. The most important finding is the strong seasonability of ice nucleation particles (INPs) with lower concentrations in the winter and spring and enhanced concentrations of INPs during the summer melt. Similar findings have also been reported by earlier campaigns (see ref. 18, 24, 25). Since this effect is of high importance for the understanding of the whole climate system of that region, I would have expected that the authors go into more details regarding this effect. Here are some questions I would like to see been answered:

A) Bioaerosol particles (1-5 μm), e.g. fungal spores, their fragments, and sub-pollen particles, are expected to travel long distances. Could long range transport bring INP rich air and precipitation from the continents into the Arctic, which is then deposited on the ice sheet?

B) Can you distinguish between INPs from the open ocean water and from the melting water of the ice?

C) Could you determine the amounts of marine algae and bacteria versus continental bacteria, fungal spores and sub pollen particles?

D) Did you check for the amount of culturable microorganisms?

E) Can you distinguish between very active and less active INPs in terms of the different types of bioaerosols: bacteria, algae, pollen, fungal spores, sub-pollen particles and sub-spore particles?

F) Did you apply spectroscopic techniques to measure the aerosols, e.g. a Wideband Integrated Bioaerosol Sensor?

G) Did you analyze the foils from the stages of your impactor by microscopy and/or spectroscopy in order to gather information of the origin of the aerosol?

H) In Fig 2 c) and d) one can see the aerosol inlet temperature and the chl-a concentration rising in parallel between April and July. However, the temperature stays high until October while the chl-a concentration falls after July. What can we learn from these trends? This is not in accordance with the aerolization of marine water but is more plausible for the aerolization of melt water from the ice.

From my point of view the here described research is very much focused on observations while explanations and considerations are partly missing. I would like to encourage the authors to clarify the following aspects:

A) What is the reason for the seasonability of the INPs activity?

B) What is the exact origin of the INPs?

C) What are the decisive lateral and vertical transport mechanisms of the INPs?

D) Did you find other field campaigns which describe similar seasonal effects of INP activity, e.g. P. Baloh et al. describe in Science of The Total Environment 2021, 800, 149442 a seasonal trend of INPs in the polar climate regions of the Alps (<https://doi.org/10.1016/j.scitotenv.2021.149442>).

From my point of view this paper should be published after substantial revisions and answering the above listed questions.

We would like to thank the reviewers for their valuable comments and feedback. We have revised the manuscript accordingly and think it has strengthened as a result. Please find our responses to reviewer comments and changes to the manuscript below in blue text. A track changes version is also included.

Reviewer #1 (Remarks to the Author):

The manuscript by Creamean et al. presents the seasonal variations in the concentration, size distribution and possible sources of ice nucleating particles (INPs) in the central Arctic based on a year-round cruise observation. The results are useful to provide cloud parameters in climate models. Whereas, in my opinion, some explanations on the results are not so sound. The influencing factors and the sources of INPs are mainly interpreted based on environmental conditions and air mass trajectories, but direct evidences (e.g., chemical tracers, microbial abundance and compositions) are lack. Direct measurements of aerosol/INP chemistry are required to more accurately identify the sources and influencing factors of INP populations in the central Arctic.

My specific comments are listed as follows:

1. Title: “aerosols that seed ice formation in central Arctic clouds”. The results of the presented study show the potential of the aerosol particles in the central Arctic that may participate in ice cloud formation rather than the aerosol particles seed ice formation in central Arctic clouds. The title should be revised to be more accurate.

Since we have measured the potential of these INPs to seed Arctic clouds, we changed the title to, “First annual cycle observations of aerosols capable of ice formation in central Arctic clouds”.

2. Line 52-54: Previous studies have reported that ash, soot or organic particles may also be ice nucleation-active, although the results are still controversial (Cozic et al., 2007, 2008; Levin et al., 2016; Umo et al., 2015; McCluskey et al., 2017; Wolf et al., 2020; Wilson et al., 2015). How about the contributions of other particle types (e.g., ash, soot or organics) to INPs in the Arctic, in addition to mineral and biological materials? Moreover, understanding the compositions of the INPs is crucial to capture the breadth of INPs (Line 59), although some studies might suggest that anthropogenic pollution does not contribute to atmospheric INPs.

Given the succinctness of Nature articles, and specifically, the introduction, we refrained from providing a comprehensive overview of INP types globally and instead focused on dominant INP types found in the Arctic. We changed the wording of this sentence to reflect that these have been observed as the dominant, not the only, types of Arctic INPs.

While sources such as ash and soot may be influential to INP populations at lower latitudes, these particles are likely more significantly aged by the time they reach the central Arctic, meaning findings from the lower midlatitudes may not be applicable here. To our knowledge, there is no evidence of ash being measured as INPs in the Arctic. The closest is INP measurements from Eyjafjallajökull in 2010, but there was no discussion on transport of INPs from this eruption to the Arctic circle (Sanchez-Marroquin et al., 2020). These sources are rare and produce poor immersion INPs (e.g., Hoyle et al., 2011; Steinke et al., 2011). “Organics” is a wide range of aerosol type that emanate from many sources including those described in the manuscript, soil dust and the ocean. While smoke has recently been postulated as a source of INPs in the upper troposphere of the Arctic (Engelmann et al., 2021), that work was based on a lidar estimation method of INPs active at cirrus-relevant temperatures and not in situ observations relevant for AMPCs. Several studies report that pollution is not a source of INPs, including in the Arctic (Wex et al., 2019). To our knowledge, no studies report in situ observational evidence of pollution as a significant source of INPs. The closest is Zhao et al. (2019) but their work is based on relationships between AOD for polluted continental aerosols from CALIOP and ice particle effective radius at cold-top convective clouds and anvil cirrus over East Asia.

- Engelmann R, Ansmann A, Ohneiser K, Griesche H, Radenz M, Hofer J, et al. Wildfire smoke, Arctic haze, and aerosol effects on mixed-phase and cirrus clouds over the North Pole region during MOSAiC: an introduction. *Atmos Chem Phys* 2021, 21(17): 13397-13423.
- Hoyle CR, Pinti V, Welti A, Zobrist B, Marcolli C, Luo B, et al. Ice nucleation properties of volcanic ash from Eyjafjallajökull. *Atmos Chem Phys* 2011, 11(18): 9911-9926.
- Sanchez-Marroquin A, Arnalds O, Baustian-Dorsi KJ, Browne J, Dagsson-Waldhauserova P, Harrison AD, et al. Iceland is an episodic source of atmospheric ice-nucleating particles relevant for mixed-phase clouds. *Science Advances* 2020, 6(26).
- Steinke I, Möhler O, Kiselev A, Niemand M, Saathoff H, Schnaiter M, et al. Ice nucleation properties of fine ash particles from the Eyjafjallajökull eruption in April 2010. *Atmos Chem Phys* 2011, 11(24): 12945-12958.
- Zhao B, Wang Y, Gu Y, Liou K-N, Jiang JH, Fan J, et al. Ice nucleation by aerosols from anthropogenic pollution. *Nature Geoscience* 2019, 12(8): 602-607.

3. Line 84-86: The sum percentages of airmasses over the pack ice, MIZ and open water or land are not 100%. What does “north of 80N (79% of the time) mean”? In addition, the authors set the initial height of backward trajectories as 12 m above the AOS (Line 63 in the supplementary), which might ineffectually estimate the trajectories of airmasses because the meteorological conditions of the very-near surface of the ocean may be inaccurate due to intense turbulences.

Thank you for pointing the percentages out; there was a typo for the percentage of time spent over open water or land.

The 12 m over the AOS inlet was chosen to represent the closest vertical level that would be realistic for airmasses influencing the sampled air, while avoiding possible local interferences from the ship infrastructure, which are not accounted for in the meteorological field data used in the trajectory analyses. Anything higher may not be representative of what was measured at the level of the inlet.

“North of 80°N” means the average trajectory latitude was > 80°N. This is the region where pack ice was present during most of the year, except for the minimum in Sep 2020. We realize this is a bit redundant from the first statistic in this sentence, so omitted it.

4. Line 94-99: Are the INP concentrations illustrated in each panel of Fig. 5 accumulated for every 2.5C interval, or newly frozen ones in each 2.5C interval? The results described here are not obviously illustrated in Fig. 5. Cold-temperature INPs during the winter were likely comparable to those during other seasons.

These were originally cumulative INP concentrations. However, we realize this is not the best way to represent them in this figure since we discuss cold- versus warm-temperature INPs, as cumulative concentration at -22.5 °C, for example, includes both INPs at that temperature and INPs at warmer temperatures. We have revised this figure (Fig. 4) to show the differential INP concentrations at each temperature. Cold-temperature INPs during winter were only comparable to late summer (not spring or fall), specifically at -20.0 and -22.5 °C.

To help corroborate the sources we discuss throughout the manuscript, we have added supporting information on INP speciation determined using treatments of heat and peroxide. These treatments are well-established techniques to determine the relative abundance of INPs that are heat-labile (i.e., proteinaceous or biological), bio-organic, and inorganic (i.e., often mineral). We added a figure showing these results at various temperatures over the course of the expedition (Fig. S9), described the methods for sample collection and processing, and report these findings throughout the manuscript when summarizing the INP sources.

We now indicate that cold-temperature INPs peaked during the winter and late summer: “Cold-temperature INPs (specifically at -20.0 and -22.5 °C) peaked during the winter and late summer. The winter peak was likely influenced by dust from continental sources²⁶⁻²⁸, as shown by the intermittent

transport of continental air masses from Siberia, Eastern Europe, and Northern Canada (Figs. S4–S6) and the presence of mostly inorganic (i.e., mineral) INPs at temperatures below $-20\text{ }^{\circ}\text{C}$ (Fig. S9). The late summer peak was likely also from lower latitude sources, as the Polarstern reached the ice-free ocean during transit (Figs. 2a and 3a).”

5. Line 102-110: The two results reported here are somewhat self-contradictory. Abundant warm-temperature INPs coincided with elevated Chl-a concentration as well as increased open water and melt pond fractions within the pack ice, whereas the authors concluded that ice-free ocean was not a significant source of the warm-temperature INPs in the summer based on the unenhanced warm-temperature INPs during the transit periods in open water. The authors inferred summer INPs were likely from biological productivity in open water sources either locally within the pack ice or from farther north over the pack ice. These indications are confusing. How can sea spray particles be produced by winds only from localized open water sources within the pack ice but not ice-free water? What are the possible mechanisms? More direct evidences on the sources of the INPs are required, e.g., chemical and microbial compositions of the Arctic aerosols.

Open water sources within the pack ice (i.e., melt ponds and leads) and ice-free ocean (i.e., open ocean south of the pack ice and MIZ) are disparate sources. While the mechanisms for sea spray generation could be similar under the same wind regimes, not all open water equates to the same type of sea spray. Open water within the pack ice and the MIZ contain a mixture of ice algal and marine microbial communities that are more densely populated due to the freshly-melted ice, while these microorganisms effectively sink in larger areas of open water that has been devoid of ice for longer periods. This process is probably best depicted in Fig. 5 of Ardyna et al. (2020; see below). Under ice (Ardyna et al., 2020) and ice edge (Perrette et al., 2011) blooms are prominent as the melt season progresses and the highest density of these microbial communities is closer to the surface during mid-summer. As the water opens completely, these blooms descend from the surface and farther away from potential exchange with the atmosphere. While some microorganisms can remain closer to the surface, the communities are not as dense as during melt onset.

Furthermore, marine INPs from pristine SSA are typically not as proficient as, say, dust (e.g., McCluskey et al., 2018a, 2018b), unless those waters are subject to shallow shelf regions or collapsing phytoplankton blooms (e.g., Creamean et al. 2019; McCluskey et al., 2018b). Thus, we would not expect ice-free open water to be a prolific INP source, especially at warm temperatures.

To distinguish between these two open water sources (local melt ponds and MIZ versus ice-free ocean), we modified the wording in this section by providing more detail about what we mean by “ice-free ocean”. We also included Ardyna et al. (2020) and added, “Additionally, as melt ponds and the MIZ develop, this exposes materials from under-ice and ice-edge algal blooms that manifest during the melt onset and extend into the summer^{30,31} and that may become airborne under appropriate wind regimes.”

*Ardyna M, Mundy CJ, Mayot N, Matthes LC, Oziel L, Horvat C, et al. Under-Ice Phytoplankton Blooms: Shedding Light on the “Invisible” Part of Arctic Primary Production. *Frontiers in Marine Science* 2020, 7.*

*McCluskey CS, Ovadnevaite J, Rinaldi M, Atkinson J, Belosi F, Ceburnis D, et al. Marine and Terrestrial Organic Ice-Nucleating Particles in Pristine Marine to Continentally Influenced Northeast Atlantic Air Masses. *Journal of Geophysical Research: Atmospheres* 2018a, 123(11): 6196-6212.*

*McCluskey CS, Hill TCJ, Sultana CM, Laskina O, Trueblood J, Santander MV, et al. A Mesocosm Double Feature: Insights into the Chemical Makeup of Marine Ice Nucleating Particles. *Journal of the Atmospheric Sciences* 2018b, 75(7): 2405-2423.*

*Perrette, M., Yool, A., Quartly, G. D. & Popova, E. E. Near-ubiquity of ice-edge blooms in the Arctic. *Biogeosciences*, 2011, 8, 515-524, doi:10.5194/bg-8-515-2011.*

6. Line 116-127: The authors stated that the results that INP concentration was highest in the pack ice at 82.1°N in the summer during MOSAiC contrasted with the results of previous studies that the contribution of INPs from ice-free ocean was significant during summertime. What could be the possible reasons?

Possible reasons could align with our explanation in comment 5 above, whereby the microbial community is denser near smaller open water sources within the pack ice and MIZ compared to the larger ice-free open ocean. This is also seasonally-dependent. For instance, the contrast from the Bigg work is likely due to their findings are from late summer, past the Jun/Jul peak in primary productivity when open water closest to the pack ice is no longer active.

Another explanation could be airmass transport variability from year-to-year and at different latitudes. We realize the wording that we used to report the findings from Hartmann et al. (2021) was likely not clear. They report their highest concentrations in the MIZ in the summer as opposed to the ice-free ocean. We changed the text to reflect this: “Hartmann et al. (2021)²⁶ reported summertime concentrations up to $1 \times 10^{-1} \text{ L}^{-1}$ at $-15 \text{ }^\circ\text{C}$ within the MIZ in July and lower concentrations over the pack ice.” While Hartmann et al. found their highest concentrations in the MIZ, their location during their summer peak in mid-July was very close to Svalbard and they observed transport from over Svalbard, MIZ, and the ice-free ocean. In contrast, our summer INP peak was at the end of Jun (Fig. 4), where we were farther north in the pack ice with most of our airmasses having spent time over the pack ice prior to arrival at Polarstern (Fig. 3, Fig. S1). Note that here, we also added a more recent publication that report INPs in the late summer pack ice: “More recently, Porter et al. (2022)⁴¹ reported INP concentrations up to 1 L^{-1} at $-15 \text{ }^\circ\text{C}$ during their Aug/Sep cruise but also observed lower concentrations in the pack ice.”

More generally, other factors that could affect the location of the highest INP concentrations relative to the pack ice could be locations of blooms, interannual variability in primary productivity, and ice thickness. However, longer-term studies would be required to identify the exact parameters that affect the maximum INP concentration location. We added the following statement to the end of this section: “Possible explanations for this contrast could be variability in time of year, airmass transport pathways, ice thickness, local lead and melt pond areal fraction, and primary productivity peak location and date; however, long-term annual studies at multiple locations within the pack ice, MIZ, and ice-free ocean would be required to statistically identify the key limiting factors that control INP concentration.”

7. Line 139-144: In autumn (October, November 2019 and September 2020), a large fraction of INPs are supermicron in size. The authors infer those INPs were sea spray aerosols. What are the efficient INP compositions in the sea spray aerosols? Microbes or biomoleculars (DeMott et al., 2016; McCluskey et al.,

2017; Wilson et al., 2015)? Why were those coarse particles not deposited onto the sea surface during long-distance transport? In addition, a minor fraction of INPs are submicron in size. What might they be?

Because we observed coarse INPs with signs of transport from over open water, and coarse SSA has been shown to be a dominant aerosol type this time of year in the Arctic (Song et al., 2021; Quinn et al., 2002), it is plausible that these INPs are either microbes like heterotrophic bacteria or organics (or other biomolecules) that are internally mixed with sea salt (McCluskey et al., 2017 & 2018; Sultana et al., 2017; Wilson et al., 2015). It is also possible ice nucleating biological materials such as phytoplankton exudates are adhered to smaller (i.e., submicron) particles, like organic particles generated from the surface microlayer (Wilson et al., 2015). The only way to truly test this would be to conduct microscopy on the INPs, which would require collection of INPs on grids, either after activating in an online continuous flow diffusion chamber or offline in an environmental microscopy chamber to get the true composition and infer the source of the INPs. However, this is challenging and was not conducted for MOSAiC. We now discuss the possible types of sea spray INPs here, based on previous work, and our treatment results (see response to comment 4).

The distance of transport during the fall was relatively close to the Polarstern (Fig. S1), with episodic transport from the MIZ. Additionally, looking at Nov 2019 for example, lead fraction within the pack ice around of Polarstern spiked as did wind speeds, indicating larger sea spray aerosol could have also originated nearby as these open water sources started to freeze up going into winter. We have revised the text to reflect possible local lead and MIZ sources.

DeMott PJ, Mason RH, McCluskey CS, Hill TCJ, Perkins RJ, Desyaterik Y, et al. Ice nucleation by particles containing long-chain fatty acids of relevance to freezing by sea spray aerosols. Environmental Science: Processes & Impacts 2018, 20(11): 1559-1569.

McCluskey CS, Hill TCJ, Sultana CM, Laskina O, Trueblood J, Santander MV, et al. A Mesocosm Double Feature: Insights into the Chemical Makeup of Marine Ice Nucleating Particles. Journal of the Atmospheric Sciences 2018, 75(7): 2405-2423.

Mitts BA, Wang X, Lucero DD, Beall CM, Deane GB, DeMott PJ, et al. Importance of Supermicron Ice Nucleating Particles in Nascent Sea Spray. Geophysical Research Letters 2021, 48(3): e2020GL089633.

Sultana CM, Al-Mashat H, Prather KA. Expanding Single Particle Mass Spectrometer Analyses for the Identification of Microbe Signatures in Sea Spray Aerosol. Analytical Chemistry 2017, 89(19): 10162-10170.

8. Line 145-150: Supermicron particles rather than submicron particles are dominant INPs as reported in previous studies (Córdoba et al., 2021; Line 135-137). Why were the results here different from the results of previous studies? Is it related to the distance or the path the airmasses transported? What are the possible contributors to the small INPs in winter?

It is probably a combination of gravitational settling, preferentially enabling the smallest particles measured to remain airborne during transit, and more efficient transport from lower latitudes during the Arctic haze. We think the small INPs are likely continental dust, based on the transport pathways, time spent over land, and the treatment results. This paragraph in the manuscript already described those possible reasons, but we added the point that INPs during this time of year had a larger contribution from inorganic (i.e., mineral) INPs here and to the beginning of the “The central Arctic INP annual cycle” section.

9. Sect. “Implications for central Arctic cloud formation”: What could be the relative contributions of locally emitted INPs and long-range transported terrestrial INPs in the central Arctic in different seasons?

Good question. Since adding the treatment data, we have provided more detail in this section about the composition of the INPs and their sources, and which are the most important each season.

10. Line 109-116, 134-137 in the supplementary: Did the size ranges measured by CPC, UHSAS and APS uniformed to aerodynamic size for comparison? The particle size cuts of the DRUM sampler should be determined by the sampling flowrate. There was a large variation in the flowrate (20-35 L min⁻¹). How could this variation influence the size ranges of the particle sampled in each stage? This will largely affect the results illustrated in Figure 5.

Because the CPC activates the aerosol into larger droplets that are then counted in total (not in discrete size bins), converting to aerodynamic diameter does not make sense. We only report the total CPC concentration throughout the year to demonstrate when larger concentrations of particles are present in the summer.

We also corrected the UHSAS data to convert optical diameter (i.e., assumed to be the true diameter) to aerodynamic diameter, using the following equation and density of 1.5 g cm⁻³ based on previous studies in both pristine and polluted locations (e.g., Kannosto et al., 2008; Zhao et al., 2017): $D_a = D_{true} \times \sqrt{1.5} = 1.22 \times D_{true}$. We included the details on this conversion in the methods section, when describing the UHSAS.

We realized we originally reported the range for all samples, including 3 outlier days during the daily DRUM sample collections. However, we changed our values to 29–34 L min⁻¹ (average flow rate of 31.6±1.6 L min⁻¹ because these are the flow rates for the samples processed for INP data. It is unlikely that a variation in this small of a range of flow rates would substantially change the size cuts of the DRUM orifices.

Kannosto J, Virtanen A, Lemmetty M, Mäkelä JM, Keskinen J, Junninen H, et al. Mode resolved density of atmospheric aerosol particles. Atmos Chem Phys 2008, 8(17): 5327-5337.

Zhao S, Yu Y, Yin D, He J. Effective Density of Submicron Aerosol Particles in a Typical Valley City, Western China. Aerosol and Air Quality Research 2017, 17(1): 1-13.

References:

Cozic, J., Verheggen, B., Mertes, S., Connolly, P., Bower, K., Petzold, A., Baltensperger, U., and Weingartner, E.: Scavenging of black carbon in mixed phase clouds at the high alpine site Jungfraujoch, Atmos. Chem. Phys., 7, 1797–1807, 2007.

Cozic, J., Mertes, S., Verheggen, B., Cziczo, D. J., Gallavardin, S. J., Walter, S., Baltensperger, U., and Weingartner, E.: Black carbon enrichment in atmospheric ice particle residuals observed in lower tropospheric mixed phase clouds, J. Geophys. Res.-Atmos., 113, D15209, 2008.

Córdoba, F., Ramírez-Romero, C., Cabrera, D., Raga, G.B., Miranda, J., Alvarez-Ospina, H., Rosas, D., Figueroa, B., Kim, J.S., Yakobi-Hancock, J., Amador, T., Gutierrez, W., García, M., Bertram, A.K., Baumgardner, D., Ladino, L.A., 2021. Measurement report: Ice nucleating abilities of biomass burning, African dust, and sea spray aerosol particles over the Yucatán Peninsula. Atmos. Chem. Phys. 21, 4453-4470.

DeMott, P.J., Hill, T.C., McCluskey, C.S., Prather, K.A., Collins, D.B., Sullivan, R.C., Ruppel, M.J., Mason, R.H., Irish, V.E., Lee, T., Hwang, C.Y., Rhee, T.S., Snider, J.R., McMeeking, G.R., Dhaniyala, S., Lewis, E.R., Wentzell, J.J., Abbatt, J., Lee, C., Sultana, C.M., Ault, A.P., Axson, J.L., Diaz Martinez, M., Venero, I., Santos-Figueroa, G., Stokes, M.D., Deane, G.B., Mayol-Bracero, O.L., Grassian, V.H., Bertram, T.H., Bertram, A.K., Moffett, B.F., Franc, G.D., 2016. Sea spray aerosol as a unique source of ice nucleating particles. Proc. Natl. Acad. Sci. USA 113, 5797-5803.

Levin, E. J. T., McMeeking, G. R., DeMott, P. J., McCluskey, C. S., Carrico, C. M., Nakao, S., Jayarathne, T., Stone, E. A., Stockwell, C. E., Yokelson, R. J., and Kreidenweis, S. M.: Ice nucleating particle emissions from biomass combustion and the potential importance of soot aerosol, J. Geophys. Res., 121, 5888–5903, 2016.

McCluskey, C.S., Hill, T.C.J., Malfatti, F., Sultana, C.M., Lee, C., Santander, M.V., Beall, C.M., Moore, K.A., Cornwell, G.C., Collins, D.B., Prather, K.A., Jayarathne, T., Stone, E.A., Azam, F., Kreidenweis, S.M., DeMott, P.J., 2017. A Dynamic Link between Ice Nucleating Particles Released in Nascent Sea Spray Aerosol and Oceanic Biological Activity during Two Mesocosm Experiments. *Journal of the Atmospheric Sciences* 74, 151-166.

Umo, N. S., Murray, B. J., Baeza-Romero, M. T., Jones, J. M., Lea-Langton, A. R., Malkin, T. L., O’Sullivan, D., Neve, L., Plane, J. M. C., and Williams, A.: Ice nucleation by combustion ash particles at conditions relevant to mixed-phase clouds, *Atmos. Chem. Phys.*, 15, 5195–5210, 2015.

Wilson, T.W., Ladino, L.A., Alpert, P.A., Breckels, M.N., Brooks, I.M., Browse, J., Burrows, S.M., Carslaw, K.S., Huffman, J.A., Judd, C., Kilhau, W.P., Mason, R.H., McFiggans, G., Miller, L.A., Nájera, J.J., Polishchuk, E., Rae, S., Schiller, C.L., Si, M., Temprado, J.V., Whale, T.F., Wong, J.P.S., Wurl, O., Yakobi-Hancock, J.D., Abbatt, J.P.D., Aller, J.Y., Bertram, A.K., Knopf, D.A., Murray, B.J., 2015. A marine biogenic source of atmospheric ice-nucleating particles. *Nature* 525, 234-238.

Wolf, M.J., Zhang, Y., Zawadowicz, M.A., Goodell, M., Froyd, K., Freney, E., Sellegri, K., Rosch, M., Cui, T., Winter, M., Lacher, L., Axisa, D., DeMott, P.J., Levin, E.J.T., Gute, E., Abbatt, J., Koss, A., Kroll, J.H., Surratt, J.D., Cziczko, D.J., 2020. A biogenic secondary organic aerosol source of cirrus ice nucleating particles. *Nat. Commun.* 11, 4834.

Reviewer #2 (Remarks to the Author):

I support publication. Long term (more than a few weeks) measurements of ice nucleating particles (INPs) are relatively rare. A seasonal cycle in the Arctic is indeed a unique dataset – all the more relevant because of the importance of ice in mixed phase clouds in the Arctic, as the authors note. Further, these measurements are size selected. As rare as documentation of a seasonal cycle of INPs is... Size selected is even more rare.

Is this the first study to report a seasonal cycle of size selected INPs? I don’t recall another one. The longest period in Mason et al. (ACP, cited in the paper) is a few months. If this is indeed the first study to show this long a time series of size selected INPs, that should be mentioned more prominently.

Thanks for pointing this out. To our knowledge, it is indeed the first report of the annual cycle of size-resolved INPs. In the abstract, we added, “Further, these are the first reported annual cycle observations of size-resolved INPs, globally.” At the end of the introduction, we added, “To our knowledge, this is the first time an annual cycle of size-resolved INPs has been reported, not just in the Arctic, but at any location globally.” We also added the following statement to the beginning of the size-resolved results section: “In fact, these are the first reported observations of a full annual cycle of size-resolved INP data anywhere globally.”

The time series of INP as a function of temperature is quite striking. My only quibble with that figure is that the scale on the ordinate axis changes with temperature. This can be quite misleading. For example, one might conclude that the concentrations for -7.5 and -10 are the same until noticing the factor of 10 difference in the scale. Note that it is MUCH MUCH easier to notice such a scale change in the review version of the paper where the figure is scaled up to roughly the size of a page. At least note this change in scale in the caption. (I’m not 25 any more... My eyesight isn’t as good as it used to be, and it would be much harder to tease this out in the final version of the article where the figures are much smaller.)

Valid point, I (Jessie) am not 25 anymore either so I understand! We increased the font size of the y-axes and noted in the caption, “Note the difference in INP concentration scales between the different temperatures.”

I also have one cautionary note to add concerning the size of the particles. Figure 5 is quite striking, and it is very interesting to me that the smaller particles make up the majority of INP in some months (January, for example). I think the argument based on transport is an interesting one, but some caution is warranted there. The shape of the particles is unknown and nonspherical particles will settle slower than spherical ones of the same volume. See Yang et al for one discussion of that. It's also worth noting that the fraction of particles in a given size range which act as INP isn't shown. Take January for example. The absolute number of particles that act as INP is dominated by the 150 to 340 nm size range but there are also many more of those particles. My back of the envelope, reading rough numbers off the graphs calculation indicates that 1 in a million particles was an INP (at -20) in both size ranges. That's an interesting number, in itself, but also an interesting perspective on the dominance of the smaller size range as INP. (This doesn't account for the fact that the larger particles are very likely to be irregularly shaped, so the aerodynamic size that's reported here is hard to convert to a surface area...)

We have now included a figure in the SI that shows the monthly ratios of INPs to total aerosol within the 4 size ranges (Fig. S8). Throughout the size-resolved section, we included text that discusses the size-resolved INPs relative to the total aerosol (the ratios). We also added the following to the "Airmass trajectory modelling" section of the methods that includes the Yang et al. (2013) reference: "Additionally, particle sphericity impacts the rate of gravitational settling¹⁸ but neither particle sphericity or gravitational settling are accounted for here, as the relevant measurements to determine particle sphericity were not made and HYSPLIT is a passive air parcel model."

In summary, the authors have presented quite an interesting dataset, worthy of publication in Nature Communications, in my opinion. I support publication.

Yang, W., Marshak, A., Kostinski, A.B. and Várnai, T., 2013. Shape-induced gravitational sorting of Saharan dust during transatlantic voyage: Evidence from CALIOP lidar depolarization measurements. Geophysical Research Letters, 40(12), pp.3281-3286.

Reviewer #3 (Remarks to the Author):

This is the first annual cycle observation of aerosols that seed ice formation in central Arctic clouds. The paper is well-written and reports interesting findings which are important for understanding the regional climate in one of the most sensible regions of the globe. The most important finding is the strong seasonability of ice nucleation particles (INPs) with lower concentrations in the winter and spring and enhanced concentrations of INPs during the summer melt. Similar findings have also been reported by earlier campaigns (see ref. 18, 24, 25). Since this effect is of high importance for the understanding of the whole climate system of that region, I would have expected that the authors go into more details regarding this effect. Here are some questions I would like to see been answered:

A) Bioaerosol particles (1-5 μm), e.g. fungal spores, their fragments, and sub-pollen particles, are expected to travel long distances. Could long range transport bring INP rich air and precipitation from the continents into the Arctic, which is then deposited on the ice sheet?

This certainly is possible. To help corroborate the sources we discuss throughout the manuscript, we have added supporting information on INP speciation determined using treatments of heat and peroxide. These treatments are well-established techniques to determine the relative abundance of INPs that are heat-labile (i.e., proteinaceous or biological), bio-organic, and inorganic (i.e., often mineral dust). We added a figure showing these results at various temperatures over the course of the expedition (Fig. S9), described the methods for sample collection and processing, and report these findings throughout the manuscript when summarizing the INP sources. However, note these data are for total aerosol samples, so not specific to 1-5 μm particles.

Evidence points to larger biological INPs being present during summer periods. Periods where transport was prominent from lower latitudes (e.g., the Arctic haze months), INPs were generally smaller and inorganic but larger INPs were still present to a smaller extent. To account for this possible minor source, we added the following to the size-resolved section when discussing winter, "However, larger INPs were still present, albeit in lower concentration, which could be attributed to dust or biological INPs that are active at relatively low temperatures compared to summer."

B) Can you distinguish between INPs from the open ocean water and from the melting water of the ice?

With the data we currently have, we cannot distinguish INPs from these sources in detail. This would require DNA sequencing and bioinformatics to determine if the general aerosol population had microbial signatures of sea ice versus marine (Arctic Ocean) algal and bacterial communities. Even then, we would have information on the total aerosol population and not the INPs alone. Determining the microbial makeup of the INPs specifically is not possible, but one could postulate based on relationships between the total aerosol microbial community and INP concentrations (e.g., if airmasses from over melt ponds contained a unique DNA signature and led to increased INP concentrations at warm temperatures). We can only use the supporting data and air mass transport pathways to hypothesize what the sources might be.

C) Could you determine the amounts of marine algae and bacteria versus continental bacteria, fungal spores and sub pollen particles?

While this would certainly be very interesting and valuable, it would require DNA sequencing and bioinformatics. Both 16S and 18S rRNA sequencing are in progress and will be published in a future study. The goal will be to analyze bacterial and eukaryotic community composition (which includes fungi and pollen) over the course of the campaign and compare to trends in the INP data. Unfortunately, amplicon sequencing only looks at relative shifts and not overall taxa amounts. Quantitative PCR may be used to estimate absolute abundance, although many samples have extremely low amounts of genomic DNA and do not amplify well. See response to the comment above.

D) Did you check for the amount of culturable microorganisms?

No, we did not. Because INPs of biological origin can be primary biological particles (culturable and non-culturable) and organic byproducts such as exudates and other macromolecules, culturing would miss a substantial part of the population of particles that could be the INPs. However, as described in comment A, we now include the treatment data which do at least provide some information on the relative abundances of biological and organic INPs.

E) Can you distinguish between very active and less active INPs in terms of the different types of bioaerosols: bacteria, algae, pollen, fungal spores, sub-pollen particles and sub-spore particles?

See response to comment A.

F) Did you apply spectroscopic techniques to measure the aerosols, e.g. a Wideband Integrated Bioaerosol Sensor?

A WIBS was deployed during MOSAiC but is currently undergoing quality control and validation. Additionally, it is subject to a future publication as a part of a graduate student's PhD work (co-author Ivo Beck) to evaluate relationships between aerosol fluorescence and INPs. We are looking forward to evaluating these potential relationships in the future.

G) Did you analyze the foils from the stages of your impactor by microscopy and/or spectroscopy in order to gather information of the origin of the aerosol?

The samples were collected on a plastic material coated with Vaseline (see the methods), thus would not be ideal for microscopic techniques. One of the MOSAiC PIs (Prof. Kerri Pratt from University of Michigan) deployed a 3-stage DRUM that contained grids for CCSEM-EDX. However, this analysis is incredibly time and personnel intensive, and will take several years before these data are available. This long wait time is also a result of pandemic-related delays in access to microscopy facilities. A handful of case studies will be evaluated as a separate project for a postdoc in Kerri's group, as a part of a newly-funded project with PI Creamean, co-I Schmale (coauthor on current manuscript), and co-I Pratt. Thus, the origin of the aerosol will be further detailed in a future publication for select cases.

H) In Fig 2 c) and d) one can see the aerosol inlet temperature and the chl-a concentration rising in parallel between April and July. However, the temperature stays high until October while the chl-a concentration falls after July. What can we learn from these trends? This is not in accordance with the aerolization of marine water but is more plausible for the aerolization of melt water from the ice.

Good point! This relationship does indeed point to the melt water from the ice, which contains sea ice algal communities, as the likely culprit for the peak of productivity and resulting enhancement in warm-temperature INPs. We added elaborated on possible algal sources in the central Arctic INP annual cycle section: "Additionally, as melt ponds and the MIZ develop, this exposes materials from under-ice and ice-edge algal blooms that manifest during the melt onset and extend into the summer^{30,31} and that may become airborne under appropriate wind regimes."

From my point of view the here described research is very much focused on observations while explanations and considerations are partly missing. I would like to encourage the authors to clarify the following aspects:

A) What is the reason for the seasonability of the INPs activity?

The reason for the seasonality is discussed throughout the results section, i.e., fall and spring = coarse SS INPs from higher winds during freeze up, winter = smaller transported INPs during Arctic haze (which is typical for this time of year), and summer = supermicron biological INPs from local open water during the peak of primary productivity. We did add some additional information throughout the results section that discuss possible sources and mechanisms that vary throughout the year.

B) What is the exact origin of the INPs?

We cannot determine the exact origin, but we do discuss possible sources based on the INP and complimentary data, and airmass trajectory analyses. We did include additional details about the possible sources based on the treatment results and previous work. For example, we included discussion on under ice algal blooms and how those may expose INP material to the atmosphere in the annual cycle section and details on types of biological materials that could be INPs in the size-resolved section.

C) What are the decisive lateral and vertical transport mechanisms of the INPs?

While the vertical and spatial distributions of the INPs are certainly important, unfortunately, such in situ measurements were not conducted during MOSAiC (e.g., balloon/UAS or aircraft measurements of INPs). We can only postulate the likely sources of the INPs based on the trajectory analyses for passive tracers (i.e., our HYSPLIT analyses). Indeed, such measurements would be crucial in the future to address this question, thus, we added a statement in the last paragraph recommending such measurements as a part of future Arctic studies.

D) Did you find other field campaigns which describe similar seasonal effects of INP activity, e.g. P. Baloh et al. describe in Science of The Total Environment 2021, 800, 149442 a seasonal trend of INPs in the polar climate regions of the Alps (<https://doi.org/10.1016/j.scitotenv.2021.149442>).

While the work by Baloh et al. is very interesting, it reports INP activity during the months of June, July, November, and May in freshwater samples from ponds and rivers at elevations ~2200 m a.s.l. in Austria.

Thus, the sources (freshwater in the mountains), microbial processes (no sea ice or ocean), sample medium (water, versus aerosol in our manuscript), and timeframe (a few days each of the four months instead of sampling continuously throughout the full year) are not quite relevant for central Arctic INP sources and emission processes over a complete sea ice annual cycle.

From my point of view this paper should be published after substantial revisions and answering the above listed questions.

REVIEWER COMMENTS

Reviewer #1 (Remarks to the Author):

The authors gave detailed responses to the reviewers' comments and revised the manuscript accordingly. The manuscript was improved greatly, however, several comments remain to be further addressed before possible publication.

1. As mentioned last time, the authors set the initial height of backward trajectories as 12 m above the AOS or 30 m ASL. At such a low height, it is too easy to be disturbed by the underlying surface near the sea surface. In addition, the vertical resolution at the near-surface (ground) of the reanalysis data downloaded from the NCEP is too low to be accurate. If the authors use WRF to run the weather field by themselves, the resolution near the surface (ground) could be improved, and theoretically 12 m is no problem. Although the results at the initial height 12 m/30 m may be similar to those at other initial heights, I also suggest using a higher altitude (e.g., 100 m or above) for calculation.

2. The authors conducted additional treatments (heating and H₂O₂ digestion) for INP test and gave further support for their inferences, as mentioned in Comment 5, 7, 8 and 9. As commented by another reviewer, and responded by the authors, the authors measured the compositions of bacterial and fungal communities in the Arctic aerosols and the fluorescence aerosol particle concentrations during the same campaign. If possible, some related results should be added to further support the inferences as mentioned last time, although the authors responded that those results will be published in future. For instance, the relative contribution of marine versus continental biological particles, the contribution of fluorescent biological aerosol particles to INPs.

Reviewer #2 (Remarks to the Author):

I support publication. The authors have addressed the concerns that I raised in my earlier review.

Reviewer #3 (Remarks to the Author):

The answers to my comments and questions are very well-written and are convincing me. However, I miss the implementation of these answers into the manuscript. The authors have indeed presented a revised manuscript on the basis of all three referee comments. However, my feeling is that the answers

and comments to the reviews, which are not accessible to the public, have not been reproduced in the manuscript with enough detail.

The manuscript should be published after these minor corrections.

We would like to thank the reviewers for their valuable comments and feedback. We have revised the manuscript accordingly and think it has strengthened as a result. Please find our responses to reviewer comments and changes to the manuscript below in blue text. A track changes version is also included.

Reviewer #1 (Remarks to the Author):

The authors gave detailed responses to the reviewers' comments and revised the manuscript accordingly. The manuscript was improved greatly, however, several comments remain to be further addressed before possible publication.

1. As mentioned last time, the authors set the initial height of backward trajectories as 12 m above the AOS or 30 m ASL. At such a low height, it is too easy to be disturbed by the underlying surface near the sea surface. In addition, the vertical resolution at the near-surface (ground) of the reanalysis data downloaded from the NCEP is too low to be accurate. If the authors use WRF to run the weather field by themselves, the resolution near the surface (ground) could be improved, and theoretically 12 m is no problem. Although the results at the initial height 12 m/30 m may be similar to those at other initial heights, I also suggest using a higher altitude (e.g., 100 m or above) for calculation.

Thank you for the additional explanation. We now include the same trajectory analyses for 3, 5, and 7 days back in time ending at 100 m AMSL. These show similar results to the 30 m AMSL trajectory analysis, as demonstrated in the new Figs. S7–13. In the “Setting the stage: State of the sea ice, ocean, and atmosphere” on pages 3–4, we now clarify that we conducted the same analyses at both 30 m and 100 m. The caption to Fig. 3 also clarifies that the results shown there are from 30 m. The 100 m figures are now shown in the SI. We added the following statement to the SI as well: “An initiation height of 100 m AMSL was also used to demonstrate that possible disturbances from the underlying surface did not change the main conclusions and to validate the results from the trajectories ending at 30 m.”

2. The authors conducted additional treatments (heating and H₂O₂ digestion) for INP test and gave further support for their inferences, as mentioned in Comment 5, 7, 8 and 9. As commented by another reviewer, and responded by the authors, the authors measured the compositions of bacterial and fungal communities in the Arctic aerosols and the fluorescence aerosol particle concentrations during the same campaign. If possible, some related results should be added to further support the inferences as mentioned last time, although the authors responded that those results will be published in future. For instance, the relative contribution of marine versus continental biological particles, the contribution of fluorescent biological aerosol particles to INPs.

Unfortunately, these results are not yet publication ready and they will be subject to the foci of two PhD theses, in addition to two future publications. The DNA and WIBS results for a full year in the central Arctic are a first, and alone warrant their own focused publications highlighting these novel measurements. However, we added the following to “The central Arctic INP annual cycle” section (page 5): “Future publications will focus on aerosol DNA and fluorescence measurements to corroborate the presence of biological aerosol in the summer.”

Reviewer #2 (Remarks to the Author):

I support publication. The authors have addressed the concerns that I raised in my earlier review.

Reviewer #3 (Remarks to the Author):

The answers to my comments and questions are very well-written and are convincing me. However, I miss the implementation of these answers into the manuscript. The authors have indeed presented a revised

manuscript on the basis of all three referee comments. However, my feeling is that the answers and comments to the reviews, which are not accessible to the public, have not been reproduced in the manuscript with enough detail.

The manuscript should be published after these minor corrections.

We have incorporated more detail from the previous review responses in the manuscript, specifically, from the responses in which we provided explanation but did not change anything or did not indicate why we did not change anything in the first revision. These include:

- *We added more detail on why open ocean is likely not a proficient source of marine INPs melt to “The central Arctic INP annual cycle” section (page 5): “This is supported by the fact that marine INPs from open ocean sea spray are typically low in concentration, unless the waters are subject to shallow shelf regions or collapsing phytoplankton blooms^{17,29-31}.”*
- *We added more explanation on the evolution of blooms during melt to “The central Arctic INP annual cycle” section (page 5): “As the melt season progresses, these open water sources contain a mixture of ice algal and marine microbial communities that are more densely populated near the surface, while these microorganisms effectively sink in larger areas of open water that has been devoid of ice for longer periods³³, and therefore, farther away from potential exchange with the atmosphere.”*
- *We added a statement about the DNA and WIBS future work in the same section on page 5: “Future publications will focus on aerosol DNA and fluorescence measurements to corroborate the presence of biological aerosol in the summer.” (see response to comment 2 by Reviewer 1). The graduate students who will be publishing these data are coauthors on the current manuscript. Note that we are not able to discuss preliminary findings from these measurements in detail since they are not yet final, and we do not want to make any premature conclusions. Note we did not include a statement about CCSEM-EDX as there are yet no data at all that exist from those samples. We did not think it was necessary to include a statement about this measurement anywhere in the manuscript since the Pratt group is not involved with this manuscript. Future publications detailing the WIBS, DNA, and CCSEM-EDX data will explore relationships with the INP data presented here.*
- *We made it clear that the purpose of the trajectory analysis is to assess possible source regions in the “Setting the stage: State of the sea ice, ocean, and atmosphere” section (page 4): “We note that the trajectory analyses do not enable us to determine the exact origin of the airmasses, but, in tandem with complementary observations, provide insight into the possible source regions of the measured INPs.”*

Generally, we did not provide lengthy explanations given the succinctness of Nature articles but added the details we thought were sufficient to address the reviewer’s concern. If there are additional details the review is specifically referring to, then please let us know and we will revisit.

REVIEWERS' COMMENTS

Reviewer #1 (Remarks to the Author):

The authors addressed the comments of the reviewers and added the related contents. I would like to recommend the publication of the presented manuscript.